# Materials to Be Used in Future Magnetic Confinement Fusion Reactors: A Review

**DOI:** 10.3390/ma15196591

**Published:** 2022-09-22

**Authors:** René Alba, Roberto Iglesias, María Ángeles Cerdeira

**Affiliations:** Department of Physics, University of Oviedo, E-33007 Oviedo, Spain

**Keywords:** PFM, structural materials, RAFM, ODS, SiC, vanadium alloys, diagnostics, ITER, DEMO, IFMIF-DONES

## Abstract

This paper presents the roadmap of the main materials to be used for ITER and DEMO class reactors as well as an overview of the most relevant innovations that have been made in recent years. The main idea in the EUROfusion development program for the FW (first wall) is the use of low-activation materials. Thus far, several candidates have been proposed: RAFM and ODS steels, SiC/SiC ceramic composites and vanadium alloys. In turn, the most relevant diagnostic systems and PFMs (plasma-facing materials) will be described, all accompanied by the corresponding justification for the selection of the materials as well as their main characteristics. Finally, an outlook will be provided on future material development activities to be carried out during the next phase of the conceptual design for DEMO, which is highly dependent on the success of the IFMIF-DONES facility, whose design, operation and objectives are also described in this paper.

## 1. Introduction

The global energy outlook is currently going through a time of crisis and uncertainty as a result of the excessive use of and dependence on fossil fuels over the last century. This is why a transition to new energy sources must be urgently addressed, with the need to improve efficiency and opt for a decarbonized mix in which nuclear energy seems likely to play a key role. When talking about nuclear energy, one tends to think of the present day technology (fission), but the reality is that a new means of energy production that will completely change the current paradigm is getting closer every day. Nuclear fusion energy offers the prospect of a safe, inexhaustible and waste-free energy source for generations to come. Despite this, it also presents certain science and engineering challenges that, so far, have been insurmountable due to the extreme conditions and plasma instabilities faced by the materials of these future reactors. The premise of this review is to try to analyze the horizon of new possibilities that the development of nuclear fusion is allowing and to contribute, as far as possible, to clarify what are and what could become the materials that will facilitate the success of ITER and in the future of DEMO.

## 2. Nuclear Fusion

Nuclear fusion is a reaction involving light atomic nuclei and nucleons, so that when two of these nuclei join to form a heavier one, energy is given off. It is the basis for the existence of stars like the Sun. This process initially requires the joining of a proton with another proton, an event known as the proton-proton chain, which was discovered in 1939 by the German physicist Hans Bethe [1].

Naturally, replicating this process on Earth requires in-depth research and development. Incidentally, there is one element of particular interest for fusion due to its simplicity and abundance, hydrogen, which is precisely the one used by the Sun. Specifically, two of its isotopes, deuterium (D) and tritium (T), are of interest.

This process is characterized as an exothermic reaction, where the nucleons must be very close (∼1 fm) for the strong nuclear interaction to unite them and thus overcome the electromagnetic repulsion, called the Coulomb barrier. It is also necessary to reach temperatures of millions of degrees for this reaction to take place. Almost as soon as physicists learned that solar energy could only be the product of nuclear fusion, they discovered, however, that the temperature at the center of our star (about 15 million degrees) is insufficient for hydrogen nuclei to actually come together at the necessary distance. So how is it possible for nuclear fusion to occur in the Sun? To explain this, we must resort to quantum mechanics and one of its most renowned concepts, the tunnel effect, that allows certain particles to overcome the Coulombian energy barrier without actually reaching its maximum value. Despite this, it is still necessary to reach enormous temperatures indeed. This is why fusion is described as a thermonuclear process.

As mentioned above, the most viable reaction for the first generations of nuclear fusion reactors is the one between deuterium (${}^{2}H$) and tritium (${}^{3}H$) (Figure 1), obtaining 17.6 MeV of energy that translates into an alpha particle or He nucleus and a fast neutron.
(1)$${}^{2}H{+}^{3}H\;\to {\;}^{4}He\;\left(3.52\;\mathrm{MeV}\right)+n\;\left(14.06\;\mathrm{MeV}\right).$$

Why use these H isotopes and not others? This is because the *cross section* (which defines the probability of success of a nuclear reaction between a target and an incident particle. Its SI unit is the barn, 1 barn = 10${}^{-28}$ m^2^) of this process is very high for relatively low temperatures.

Another fundamental reason is the availability of these isotopes. D can be easily found in seawater (an estimated 30 g/m^3^). T, on the other hand, is a radioactive and unstable element (decay constant $T_{1/2}=12.3$ years) which is produced naturally in small quantities when cosmic rays (98% protons) strike the H atoms present in the atmosphere. It can also be obtained as a product in CANDU NPP [2].

There are approximately 40 kilograms of T on the planet, so it is vital to find an alternative method to reproduce it on a large scale.
(2)$$\begin{array}{cc} {}^{6}Li+n & {=}^{4}He{+}^{3}H+4.86\;\mathrm{MeV}\\ {}^{7}Li+n & {=}^{4}He{+}^{3}H+n-2.5\;\mathrm{MeV}. \end{array}$$

T breeding consists of irradiating a Li blanket with the neutrons produced in the fusion reaction itself (see Equation (Equation 2)). This generates the T that will supply the reactor. Definitely, this is one of the greatest technological challenges that ITER and its related projects will have to face.

On the other hand, natural lithium (92.5% ^7^Li y 7.5% ^6^Li) is an abundant element in the earth’s crust (30 ppm) and is found in lower concentrations in the sea. The thickness of the blanket is large enough (∼m) to slow down the neutrons produced by the fusion reactions. Upon impacting with the walls, their energy is transferred in the form of heat. This heats water that turns into steam, which is then used to turn a turbine and hence generate electricity. To get an idea of the efficiency of this process, the use of 1 kg of D-T has the energy equivalence of 8000 tons of oil.

The breeding blanket (BB) is one of the most complex and important components of future fusion reactors, as it is not only responsible for the extraction of energy but also for T breeding in order to have a self-sufficient facility (tritium breeding ratio, TBR >1) [2,3]. This parameter is defined as the average number of T atoms bred per T atom burned. It should be higher than 1.15 in order to take into account T losses which cannot be avoided in a real fusion reactor [4].

Tritium transport modelling allows experts to predict how this element will move towards the systems that have to recover it in order to refuel the plasma. In doing so, two fundamental considerations must be taken into account. Firstly, tritium does not simply go from point A to B, as it is a gas that diffuses easily. This can happen especially at high temperatures, as it can enter and mix with materials in pipes, valves and other components along the way. Secondly, tritium is radioactive, so it is of great interest in terms of nuclear safety and radiological protection to know where it can accumulate.

There are two options for the commercial development of nuclear fusion, magnetic or inertial confinement. This review will look at the former, as it has become the more advanced option with a higher probability of success.

## 3. ITER: The Project That Might Change the Future

### 3.1. What Is ITER?

ITER is the most ambitious energy project in the world today and is located in the town of Cadarache in southern France (Figure 2). Up to 35 *nations* (the 27 countries of the EU together with Switzerland, the United Kingdom, China, India, Japan, Korea, Russia and the United States) are collaborating to build the world’s largest *Tokamak* (Axisymmetric toroidal chamber characterized by a large toroidal magnetic field, moderate plasma pressure and relatively small toroidal current), a magnetic confinement fusion device designed to demonstrate the viability of fusion as a large-scale, carbon-free energy source.

In turn, the technologies, materials and physical regimes required for large-scale commercial electricity production will be tested.

Thousands of engineers and scientists have contributed to the design of ITER since the idea of a joint international fusion experiment was first launched in 1985. The participating members have committed themselves over a period of about 40 years to build and operate the experimental device until fusion reaches a point at which DEMO can be launched.

### 3.2. What Are the Main Objectives?

The amount of fusion energy a tokamak is capable of producing is a direct result of the number of fusion reactions taking place inside it. The larger the vessel, the greater the plasma volume and thus the greater the potential fusion energy. This has its trade-offs in terms of cost, which is usually the case in projects based on economies of scale, typical of traditional nuclear energy production. ITER has been specifically designed to:Produce 500 MW of fusion energy.Demonstrate the safety features of a fusion device.Test the reproduction of T with TBM (Test Blanket Modules).Demonstrate the integrated operation of technologies for a fusion plant.Achieve a D-T plasma in which the reaction is maintained by internal heating [6,7].

### 3.3. When Will the First Plasma Be Obtained?

The first ITER plasma is scheduled for December 2025. As of 31 July 2022, 77.1% of the work required for the first plasma has been completed [5].

Beyond its symbolic importance, the first plasma will also be a litmus test for the project, as it will be the first occasion to verify the correct alignment of the machine’s magnetic fields as well as the correct functioning of key systems (vacuum vessel, magnets, and critical plant systems).

The first plasmas will use H, He or a mixture of both. This is why the initial processes do not require a D-T fuel. Since many of the heating systems are optimized for D-T type plasmas in order to achieve *H-mode* (a high plasma confinement operating regime which is reached when a certain heating threshold is exceeded [8]), they will operate at a reduced intensity that will gradually increase over the years. The first low-power H-plasma, which will last a few milliseconds, will be followed by other “shots” of higher power and longer duration. Finally, the first production of D-T fusion energy will take place during the nuclear phase of the machine, expected around 2035 [9].

### 3.4. Materials Design Requirements

For the range of expected operating conditions (including possible accident scenarios) in ITER and with even greater relevance to DEMO, a qualified database must be generated to demonstrate that candidate materials meet a number of indispensable design requirements. Some of these are:Good strength, ductility and toughness.Radiation resistance and nonactivation properties after irradiation.Resistance to creep rupture, fatigue cracking and creep-fatigue interactions.Good resistance of mechanical and physical properties against He embrittlement.Acceptable chemical compatibility and corrosion resistance with fusion-specific breeder materials (e.g., Be, LiPb) [10].

## 4. Plasma Facing Materials

One of the most sensitive processes taking place in a nuclear fusion reactor is the interaction with the hot plasma, which is at temperatures higher than the core of the sun. PFMs and PFCs are those materials/components that cover almost the entire internal surface of the VV and represent the interface between the plasma and the rest of the tokamak. They are part of two systems: the blanket (which includes the FW) and the divertor, which occupy areas of 610 and 140 m^2^, respectively [11].

The lifetime of a PFM is limited from ∼100 dpa (a dpa, displacements per atom, is the number of times that an atom is displaced a given fluence. It is a unit for quantifying irradiation damage that is strongly dependent on the material in question). The plasma-walls interaction processes are associated with thermal loads of up to 20 MW/m^2^ in continuous thermal periods and can reach the GW/m^2^ range when an ELM occurs. ITER will not only seek to demonstrate the feasibility of the D-T process but will also be the first test device for PFMs and PFCs in extreme radiation scenarios. Some of the most serious damaging mechanisms to be considered in these materials are:(i)T retention.(ii)H-bubble induced fragility.(iii)High velocity impacts of dust particles in the PFM.(iv)Possible degradation, transmutation and activation.(v)Thermally induced defects due to cracking and melting of PFM.(vi)Thermal fatigue damage produced in the joints between the PFM and the heat sink [12,13].

Eighty percent of the enormous amounts of heat and energy that will be produced in the fusion process will escape in the form of fast neutrons (14.1 MeV). Since these particles have no charge, they cannot be redirected by means of a magnetic field to a specific location. This has been one of the greatest engineering challenges from the outset, as the entire FW will be exposed to an intense bombardment of highly energetic neutrons; therefore, the components that will face the plasma must meet several indispensable design criteria:Be strong enough to withstand such high radiation and temperature. The material chosen should have good thermal conductivity to easily evacuate heat, but at the same time cannot be readily activated, as the components are expected to last at least 20 years before being replaced.Be capable of effectively dissipating such heat, which, recovered through a cold water circuit, will be the heat that will generate electrical power in a realistic NPP. For a material to be considered as a potential PFM component, good compatibility with the hot fusion plasma, i.e., a low atomic number, Z, is a must, as well as an excellent *sputter* (a physical process in which atoms in a solid-state (target) are released and pass into the gas phase by bombardment with energetic ions) resistance.Alternatively, tokamaks with high-Z materials such as tungsten (W) must be operated in such a way as to guarantee that the net impurity influx into the plasma should be so low that the critical impurity concentration is not exceeded. This is due to the fact that since this material has a high Z, it cannot be completely ionized, causing some of its electrons to remain free and radiate energy, thus cooling the plasma [12,14,15].

A brief analysis of the loads that the PFCs will undergo in the upcoming fusion experimental projects is presented in Table 1. It can be seen that the step between ITER and DEMO is much larger than between DEMO and *PROTO* (DEMO’s successor, expected to become the first commercial nuclear fusion reactor after 2050), hence the need for an intermediate materials testing facility between ITER and DEMO.

The neutron load is the energy of the 14-MeV neutrons from the D–T reaction which pass through the FW. Although they are not deposited in the FW, they can damage it. The neutron load accumulated over the lifetime of each project is the parameter that really matters. This is substantially larger for a reactor than it is for ITER because a reactor should last for roughly 15 years before it needs to be upgraded. ITER is only an experiment. The longer the material is exposed to a neutron flux, the more frequently one of its atoms will be knocked out of place by a neutron. After many dpa events, the material will swell or shrink and become so brittle as to be useless [14].

By 2050, DEMO is estimated to have the capacity to supply 100 MW of net power to the grid and operate on a closed fuel cycle. However, for this to happen, the materials need to overcome much tougher conditions than those they will face in ITER. The need to include sufficient flexibility in the design of DEMO to accommodate improvements in plasma performance and design of core components is indispensable [17].

The main aims of DEMO are to:Achieve T self-sufficiency.Solve all physical and technical issues related to the plant and demonstrate reactor-related technology.Achieve adequate availability/reliability operation over a *reasonable time span* (while ITER is expected to work with 400 s pulses and a long dwell time, DEMO will work with long pulses (>2 h) or even at a steady state) [17,18].

Materials for the DEMO reactor must be chosen considering the high doses of irradiation produced by neutrons with a thermonuclear reaction energy spectrum and the very high heat load on the inner wall of the chamber. This can lead to significant in-vessel material damage. It will be necessary to develop and test new materials for constructing the DEMO thermonuclear reactor and solve issues related to their commercial-scale manufacture [15,17].

The blanket supposed to be used in ITER is in fact unsuitable for the future DEMO thermonuclear reactor. The envisaged materials can only withstand a small neutron flux and the exit coolant temperature is too low to ensure efficient power generation. The next step after ITER is to elaborate the design of DEMO and a thermonuclear power plant. Their linear dimensions will be about 50% larger than those of ITER, and their fusion power will be 5 and 7 times higher, respectively [15].

It appears that pilot plants and reactors may experience rates of net erosion and deposition of PFC material in the range of 10^3^–10^4^ kg/year, values well above those expected in ITER. The deposition of such massive quantities of material has the potential to interfere with pilot plant and reactor operation and to seriously compromise the safety of the DT cycle. For example, elevated dust levels due to exfoliated and detached deposits can lead to a high risk of dust explosion. Other adverse effects due to the accumulation of unwanted eroded material at critical locations could result in the appearance of cracks in the cooling channels due to thermal stress [19].

The DEMO design and R&D activities will benefit largely from the strongest experimental supporting evidence that will be gained from the design, construction and operation of ITER. Due to the *differences* (in terms of size and, especially, in terms of ITER’s wider mission) between the two devices, not all ITER solutions are directly applicable to DEMO [18,20].

### 4.1. Tungsten

Nowadays, tungsten is considered the most efficient material for components facing high heat flux, mainly due to its high melting point ($T=3422{\;}^{{}^{\circ}}$C), good thermal conductivity (160 W/m·K), excellent high temperature stability and low T retention [12,21].

However, major concerns regarding the use of W in fusion reactor applications include its inherent brittleness at low temperature and the embrittlement due to recrystallization and neutron irradiation. To overcome these drawbacks, several efforts have been made to modify W through grain refining, alloying, dispersion of secondary phases, and formation of composites.

Although W and CFCs were initially considered as the most promising PFMs, the fact is that at the end of 2013, the decision was made to discard CFCs due to their tendency to retain T and to opt for a fully tungsten-armoured divertor [22].

A *single null divertor* (characterised by toroidal symmetry and one X-point or “null”) is to be installed in the bottom area of the VV of the ITER tokamak. It will extract the heat and ash produced in the fusion reaction, minimize plasma contamination and protect the surrounding walls from thermal and neutron fluxes.

Specifically, the divertor will consist of 54 divertor cassette assemblies (CAs) operated by remote handling (Figure 3). Each of these CAs includes a cassette body (CB) and three PFCs, namely the internal and external vertical targets (IVT and OVT) and the dome. In addition, each of these modules will house diagnostic components for plasma control, evaluation and optimization [11,12].

The IVTs and OVTs are placed at the intersection of the magnetic field lines where the particle bombardment will be particularly intense in ITER. The heat flux to which these components are subjected is estimated to be between 10 and 20 MW/m^2^. Materials and cooling methods that cannot be used in the FW may be used in the divertor, mainly due to the presence of coils located at the bottom of the chamber that bend the outermost field lines to enter the divertor [14].

#### 4.1.1. ITER Design

During the last decades, several PFC designs have been developed with W. The most efficient are the monoblock and flat-tile types.

These models (Figure 4) consist of modules that have been machined from a PFM and attached to a water-cooled heat sink made of a metalic alloy. The joints between the PFM and the heat sink must acquire very high mechanical strength to tolerate the high temperatures and keep the modules uniformly in position. Each of these is equipped with a cylindrical hole necessary for the junction between the PFM and the heat sink—usually made of CuCrZr.

Despite its good performance, the flat tile design (Figure 4b) presents the possibility of local overheating of the shielding plate due to the incidence of plasma particles. The loss of even a single tile is considered a rather serious event, as it would lead to the degradation of joints in the adjacent tiles (so-called cascade failure) [12,23].

Therefore, it has been decided that the ITER divertor will be completely made of monoblock W due to its greater robustness versus possible accident conditions. Thus, both IVTs and OVTs will be completely covered with this water-cooled material (Figure 5a) [24].

#### 4.1.2. Radiation Effects

The much-feared effect that radiation can have on the divertor components in ITER is shown in Figure 6.

In this case, quite serious macroscopic damage resulting from cyclic thermal loads simulating ELMs can be observed. Specifically, 10^5^ pulses with a heat flux of 12 MW/m^2^ have been applied to a W sample preheated to 700 °C. It shows a very intense degradation due to the formation of a dense network of cracks on the surface. The 10^5^ pulses of this experiment correspond to an operating time of only 10 standard plasma discharges in ITER. It points out the potential danger that these pulses can represent to this type of materials [12].

#### 4.1.3. Smart Alloys

Due to its excellent properties, W has also been chosen as a prime PFM candidate for DEMO. However, certain accidental models have revealed several drawbacks related to the use of pure W. Those are its intrinsic low temperature brittleness, neutron-induced embrittlement and recrystallization resistance [21]. To prevent this, new modalities have been developed and tested under fusion-relevant conditions. These include the modification of its granular structure and the combination with several alloying elements and compounds (Mo, Ti, Y_2_O_3_, …) to increase strength and recrystallization resistance. The latter are the so-called *smart alloys* (which automatically adapt their properties to the environment [25]) (SA) [12,23].

One of the major concerns of using pure W in DEMO is related to its behavior under a LOCA with air ingress in the VV. Under those circumstances, the temperature of the tungsten cladding could reach 1000 °C and remain at such a high level for several weeks. Tungsten oxidizes and radioactive, neutron-activated tungsten oxide sublimates into the environment at such a high temperature.

During an accident, the remaining alloying elements in the bulk will diffuse to the surface and form their own oxides, protecting tungsten from oxidation and subsequent sublimation into the atmosphere. Recent studies have demonstrated the benefits of systems based on W-Cr-Y, which are produced by *mechanical alloying* (a ball milling process where a powder mixture placed in the ball mill is subjected to a high-energy collision from the balls. The process is usually carried out in an inert atmosphere [26] ) (MA) and compacted by FAST. These SA have demonstrated very high oxidation resistance and contain Cr as an oxidizing alloying element and Y as an active element stabilizing and regulating the chromium transport in the alloy system [27,28].

Despite all their development, there are still open questions in both the understanding of the physics and the technological development of SA systems. The role of Y in the stabilization of a W-Cr solid solution still needs to be further understood as well as the technology of joining SAs with the corresponding structural materials. At the same time, the scale of fabrication at the industrial level needs to be improved. Finally, it is of vital importance to carry out a thorough evaluation of the effect of neutrons and impurities as well as transmutation on alloy performance [28].

#### 4.1.4. Tungsten Fiber Reinforced Tungsten (W*_f_*/W)

The intrinsic brittleness of W is of great concern during possible transients with high heat loads. To reduce this brittleness, numerous procedures have been investigated to increase the toughness of the material. However, traditional intrinsic toughening methods present limitations for applications in melting environments, where high-temperature recrystallization phenomena can occur, causing severe internal damage.

To increase *fracture toughness* (the resistance of brittle materials to the propagation of flaws under an applied stress) and thus improve the intrinsic brittleness of W, tungsten fiber-reinforced tungsten composites (W*_f_*/W) are being developed for use in the divertor of future fusion reactors.

Thus far, two main fabrication approaches have been established: powder metallurgy (PM) and chemical vapor deposition (CVD) processes (Figure 7). For both cases, improved mechanical properties have been demonstrated. Generally, W*_f_*/W composites created by CVD contain 150 μm diameter unidirectional tungsten fibers coated by an interface and embedded in a W matrix [23,29,30,31].
(3)$$WF_{6}+3H_{2}⟶W+6HF$$

The CVD process (see Equation (Equation 3)) consists of applying an interface layer (e.g., Y_2_O_3_) to the fibers, exposing the composite to WF_6_ and H_2_ at temperatures between 573 and 1073 K. The fibers and CVD matrix have a major influence on the microstructure, potentially leading W*_f_*/W composites to present different properties to those of pure W material when eventually exposed to a fusion environment. Despite the fact that certain fabrication aspects still need to be further investigated, CVD is potentially one of the most cost-effective processes due to its fast deposition rate and high mass production from a reduced amount of material [31,32].

Currently, the role of K-doping in W fibers is being studied, since it has been shown to delay heat exposure-induced embrittlement at least up to 1600 °C, although a strong reduction in fiber strength has also been observed in tests conducted at elevated temperatures [33].

This doped W contains nanobubbles (∼nm) that include K atoms (∼ppm) dispersed mainly at *grain boundaries* (a 2D defect in a crystalline structure that tends to reduce the electrical and thermal conductivity of the material) (GB). Because K bubbles hinder the movement of these boundaries and dislocations, they are able to improve thermal shock resistance and mechanical properties at high temperature as well as prevent recrystallization. In addition, it is expected that the embrittlement induced by neutron irradiation can be suppressed since it contains numerous GB, which act as sinks for the defects produced. On the other hand, the addition of rhenium (Re) is also considered as another very promising procedure, namely as a solid solution alloy reinforcement [34,35].

These materials have been shown to overcome the low temperature brittleness of W. However, their main problem is industrial scale-up, which requires more effort over time and has led to the decision not to consider this material for the DEMO start-up application but to treat it as a high-potential material for further applications (e.g., PROTO) [23].

### 4.2. Beryllium

Beryllium (Be) has been on the candidate list as a PFM since the late 1980s. With the decision to use Be as the material for the ITER FW, research on its conditions and aspects most relevant to fusion has accelerated [36,37].

During ITER operation, the FW coating will be subjected to cyclic thermal loads, resulting in fatigue loads that can trigger melting, cracking, evaporation, and surface erosion [38].

Be has been selected due to its low atomic number (Z=4), which minimizes radiation losses of the sputtered atoms in the plasma (that depend on $Z^{2}$), its good thermal conductivity and its oxygen uptake capacity that contributes to maintaining a high level of plasma purity [11,12]. Beyond ITER, the three Be grades that are considered candidates for the FW of a future fusion reactor are:CN-G01 (China).S-65 VHP (USA).TGP-56PS (Russia).

These differ mainly by chemical composition, PM process used or compression method [38].

#### 4.2.1. Prototype for Use in ITER

A total of 440 panels (FWP) will provide a protective barrier for all systems beyond the VV. To get an idea of the technological challenge and the magnitude entailed by the ITER project, it is sufficient to analyze the temperature gradient existing between the hot plasma ($150\times 10^{6}$ °C) and the superconducting coils ($-269{\;}^{{}^{\circ}}$C) that will confine it, separated by a mere six meters.

In essence, the panels are made of a 6–10 mm layer of beryllium bonded to a copper alloy heat sink mounted on a *316L stainless steel* (austenitic steel composed of Cr-Ni-Mo and with a low C content) structure (Figure 8). Europe will be responsible for the production of the first 215 FWPs, while China and Russia will provide the rest of them [39,40].

In turn, Be is one of the reference neutron multipliers for the various TBM designs and is used in the form of pebble-beds. The process at the most advanced stage of development is the rotating electrode process (REP), which allows the fabrication of pebble-beds of typically 1 mm [11].

#### 4.2.2. Beryllides

Despite being considered one of the most promising materials, the main drawbacks of beryllium’s application as PFM are its relatively low melting point (T${}_{f}=1278{\;}^{{}^{\circ}}$C) as well as its high toxicity [11,12].

Beryllium intermetallic compounds (also called beryllides) such as Be_12_Ti, Be_12_V y Be_12_Zr are the most promising advanced neutron multipliers for DEMO, specifically for HCPB design [41,42].

The main consequence of neutron irradiation is significant He and T production, resulting in swelling and loss of strength-related properties of beryllium composites. DEMO-oriented R&D has focused on beryllides, as they promise to offer improved long-term material performance, as well as resulting in a much lower H production rate compared to pure Be [11].

Preliminary studies on the thermal desorption of He and T from titanium beryllium (Be_12_Ti) have shown that this material has a much lower retention tendency, in addition to having a higher melting point. Some of its strengths are:Swelling as a result of exposure to neutron irradiation occurs to a lesser extent.It has a higher melting point (1593 °C), lower activation and higher corrosion resistance.It has a sufficiently high beryllium content (69.3 wt. %) for efficient neutron multiplication/moderation [41,42,43].

All these advantages have opened the door to more extensive studies of the nuclear, physical and mechanical properties of this material with the possibility of further use in nuclear technology and high-temperature instrumentation.

### 4.3. Diamond

Due to its high Z, the presence of W in the interaction with the plasma significantly affects its stability. On the other hand, diamond has a low Z, has excellent thermal properties and thanks to its sp3 structure has a low T retention rate. For all these reasons and due to its outstanding thermal conductivity, diamond can be doped into the tungsten matrix to reduce the damage on the material.

One of the approaches that have been proposed has been to form a W/diamond composite material via SPS, which would allow the thermal conductivity of pure W to be improved. Another approach is to form diamond films via MWCVD, which, due to its chemical purity and perfect adhesive property, would allow to improve the PFM properties under high thermal loading conditions.

Both the interfacial bonding and thermal conductivity of the composite with a W-coated layer are strengthened compared to uncoated composites. The volume fraction of diamond particles in the composites is around 10–50%, and an 18% increase in thermal conductivity is achieved over pure W [44,45].

Figure 9 shows the microstructure of the fracture surfaces of diamond particle-based composites. As can be seen, for the uncoated composites, more obvious cracks appear between the diamond particles and the tungsten matrix. For the coated composites, the size and number of cracks are significantly reduced [45].

Consequently, the addition of diamond to W facilitates the manufacture of materials with outstanding strength and toughness at temperatures above 1200 °C. These composites are considered valid candidates for the FW of future fusion reactors, as they would combine excellent thermal creep corrosion toughness and resistance to neutron radiation damage [46].

It may seem that diamond doping is not an economically attractive option, but it should be borne in mind that nuclear fusion holds promise of becoming a highly safe, efficient and waste-free energy source. In order to achieve these goals, it is deemed necessary to work with materials (e.g., diamond) and technologies (e.g., the TBM program) that allow these promises to be fulfilled.

It should be noticed that the estimated budget for the construction of ITER exceeds 25 billion euros [47], which is a clear indication that no expense is being spared. Diamond is just one of many materials bringing in such beneficial properties that will provide a full return on investment from its use.

However, significant efforts are also being made to develop diamond-like carbon (DLC). This material is seen as a potential low-cost substitute for diamond in certain applications, but little is known about the temperature range over which its desirable properties are maintained. DLC coatings exist in several different forms of amorphous carbon materials that display some of the unique properties of diamond. These coatings can be amorphous, more or less flexible, hard and strong according to the composition and processing method required. Film formation can be obtained by deposition (e.g., ion beam, sputter or RF plasma) [48].

Both DLC and doped DLC films have shown attractive properties, including high hardness, low coefficient of friction or high thermal conductivity. For some applications, adherent thick DLC coatings (e.g., ∼10 μm) are desired for providing long-term durability and reliability in harsh working environments (such as those of a fusion reactor) [49,50]. By varying the production conditions, such as the bias voltage, the physical properties of the DLC can be changed to obtain coatings as hard as diamond or as soft as graphite [51].

This material has been tested for several functions in ITER. Firstly, it has been used as coating for the solid lubricant for the transmission gears of the ITER blanket maintenance equipment, thus replacing oil lubricant [51]. It was also chosen to perform qualification tests for CMS (cold mass support) sliding pads [52], where it proved to be potentially suitable as it could make the sliding interface of CMS meet all functional requirements of the CFT (cryostat feedthrough) feeder system.

Finally, it has also been used in the port plug handling system. The purpose of this system is to insert and remove the ITER port plugs installed in the equatorial and upper levels of the tokamak. Since activation of these can occur, the contamination levels prevent manual access, so their safe removal is ensured by the cask and plug remote handling system (CPRHS) between the buildings.

This handling process has been reproduced on a physical scale mock-up in which the test plug is equipped with a set of aluminium-bronze (DLC-coated) features [53].

## 5. Structural Materials

It is clear that in order to achieve projects of the nature of ITER or DEMO it is necessary to develop materials to their maximum potential, with the clear objective of withstanding the extreme conditions of temperature, irradiation damage and production of transmutation elements.

According to L. Malerba et al. [54], a structural material is one that is manufactured for the purpose of withstanding large amounts of stress, whether its origin is mechanical, thermal, vibrational, etc. These materials can be divided into two types:*Replaceable:* Designed to be relatively easy to remove from the reactor. An example would be the fuel assemblies in a nuclear power plant.*Non-replaceable:* They constitute the main structure of the reactor, so they are designed to mitigate the greatest possible degradation caused by external agents. An example would be the FW components.

It is evident that there is a strong overlap with nuclear fission materials research. However, fusion materials present a few additional challenges. The first of these is the large amount of He that is produced, both in the D-T fusion reaction and by transmutation reactions in the structure. These He bubbles that form at vacancies and GB cause swelling and embrittlement, which extensively degrades the materials. The second effect that is unique to fusion reactors is associated with the 14 MeV fast neutron. This high-energy particle penetrates deep into the structure and collides with the lattice atoms, creating numerous defects in the material. The accumulation of this damage in the structural and diagnostic materials is one of the main headaches in the design of this type of reactors [55].

Structural materials consist of crystals that adopt certain arrangements in some atoms of their lattice. Metals and alloys usually consist of regions with many crystals called grains, whose boundaries are the aforementioned GB. Ionizing radiation gives off more or less energy to the material depending on the type and energy of the particle and the medium in which it is found. This radiation-material interaction is what generates the defects, which depend directly on the initial defects of the unirradiated sample. The origin of the atomistic defects is given by:(i)Transmutation reactions.(ii)Atomic displacements due to nuclear stopping power.(iii)Ionization and excitation due to electronic stopping power.

In addition to changing the original composition of the material, the transmutation process generates H or He, elements that tend to be introduced into the cavities or voids (aggregates of vacancies). These cavities can end up coalescing, thus forming linear defects (dislocations) that eventually propagate, producing cracks or bubbles on the surface that will lead to the fracture of the material.

This forces the material to change dimensions, an effect known as swelling, which is extremely detrimental to the structural integrity of the material. The most relevant material properties are determined by the crystal defects. These are differentiated between:Point defects (e.g., Frenkel pair).Linear defects (e.g., dislocations).Planar defects (e.g., stacking faults and GB).

Point defects are of paramount importance for understanding irradiation damage and thermal properties. The movement of dislocations describes plastic deformation and is therefore key to understanding irradiation-induced changes in mechanical properties. For their part, GB are regions to which impurities can diffuse, hence the need to know their location [56].

The structural materials that make up the cooling pipes of the BB of the reactor, responsible for T production, electrical generation and radiation protection, will be subject to a severe operating environment, with damage from fast neutron irradiation, high temperature and high stress. Requirements for these structural materials include low activation, good compatibility with different coolants, resistance to irradiation and high temperatures, among others.

Thus far, three main candidates for low activation structural materials have been proposed for the FW. These are —in order of relevance— RAFM steels, SiC/SiC ceramic composites and vanadium alloys.

### 5.1. Reduced Activation Ferritic Martensitic Steels

For a conventional nuclear reaction, it is common to use stainless steel composites, but in order to withstand the extreme conditions of fusion reactions, they must reach a higher level of design. The neutrons bombarding the structure of the materials can lead to their activation, which is why low-activation materials (that do not result in long-lived radioactive isotopes) must be used. This implies that their chemical composition should be based on elements such as Fe, V, Ti, W or Ta, among others [57,58].

Two reduced-activation ferritic/martensitic steels (RAFM) have been designed for this purpose: EUROFER (Europe) and F82H (Japan). They contain the following iron additives, which make up the remaining percentages (Table 2):

These are considered the reference structural materials because they have already reached their “technical maturity”; in other words, a wide experience in terms of their manufacturing and processing methodologies has been gathered.

Unlike fission products, these steels are non-volatile and can be reused after storage for a period of 50 to 100 years. The amount of swelling they can undergo under neutron bombardment is much less than for conventional stainless steel. As with other materials, their brittleness is due to the He and H bubbles trapped in the compound in question [14].

#### 5.1.1. Test Blanket Module Program

The ITER TBM program is one of the most ambitious projects to be undertaken and plays an essential role in the design and construction of DEMO. Its objective is to develop the design that will allow to reproduce T in an efficient and safe way, while extracting heat from the blanket to generate electricity. Therefore, it is of vital importance to acquire all data and information related to the TBS (test blanket system) to provide the basis for the design, fabrication and operation of DEMO and subsequent fusion reactors [11].

From a technical point of view, the TBS are located in two equatorial ports that allow four of these TBS to operate simultaneously. Initially, they were going to be implemented in 3 ports, allowing the operation of 6 designs at the same time [11]. However, a reconfiguration was undertaken due to the need to reallocate space on the tokamak that had arisen because of initial space limitations and integration issues.

The selection process for the four designs that will be part of the initial ITER configuration is currently underway, with one possible option involving two water-cooled TBS and two helium-cooled TBS, although these will not start operating until the last non-nuclear phase of ITER [59]. All TBM designs proposed for testing in ITER use RAFM as the structural material for the reasons below:It ensures that the BB produces very limited volumes of high-level radioactive waste, thereby seeking public acceptance of nuclear fusion.It is currently the only type of material that presents the necessary structural properties and is able to meet, within the timeframe foreseen for the construction of DEMO, the necessary operational requirements. It has a good overall balance of the required mechanical properties (ductility, fracture toughness, creep or fatigue resistance), and there is extensive industrial manufacturing experience. Moreover, its optimized Cr content (8–9 wt. %) minimizes radiation-induced DBTT [11,23,60].

Four of the different designs shown in Table 3 will be part of Figure 10, which shows an overview of the 3D model of the four TBS and their associated infrastructures. Each TBS is functionally independent of the others.

Various concepts of breeding blankets are studied, with the liquid Pb–Li eutectic alloy (i.e., Pb16Li) being one of the most promising ones (used in DCLL, HCLL and WCLL). Particularly, this eutectic composition has been chosen due to its low melting temperature as compared to other Pb-Li compositions [61].

These three DEMO BBs use EUROFER as structural material, and the eutectic Pb–15.7Li enriched at 90% in ^6^Li as a breeder, neutron multiplier, and tritium carrier. T is produced inside the VV in the lead–lithium eutectic, transferred outside the VV by the eutectic alloy flow, and then extracted by the TERS (tritium extraction and removal system) [62].

This system is of vital importance as it allows the T generated in the blanket to be recovered through a loop through which the PbLi circulates. Thanks to this, the T is routed to the tritium plant to finally be re-injected into the plasma. A new facility, CLIPPER, is being constructed at CIEMAT to investigate this extraction. The design of a PbLi loop to perform experiments on H extraction from the liquid metal is presented in [63].

Nevertheless, if liquid PbLi is used, some issues arise, such as liquid metal corrosion, the behaviour of He and T in the liquid PbLi or the effects of magnetic fields in the fluid mechanics. There are still some unanswered questions, and different designs have been proposed. One of the key aspects of the future operation of liquid metal-based BBs is the eutectic composition of the PbLi alloy. There is a discrepancy on the exact eutectic point, varying the Li content from 15 at% to 17 at%. The presence of impurities and the content of this element can have a major impact on experimental activities. A crucial point is how it affects neutronic calculations on the TBR. It is assumed that PbLi does not remove any thermal power, remaining isothermal at a temperature of about ∼330 °C [4,61].

As a function of various parameters of the Pb16Li alloy (e.g., mass flow rate, temperature or pressure), the He generated will leave the breeder blanket in the form of dissolved gas, or, if the solubility limit is exceeded, as a gas bubble within Pb16Li. The amount of He bubbles generated in different DEMO-like blanket designs is estimated to be about 10–40 mL/h, which can accumulate in the system. Studies such as [64] provide valuable data for the design of liquid metal-based BB and further fundamental understanding of the inherent complexity of bubble behaviour inside liquid metals.

From all these designs and options, experts will decide which TBMs will be used in ITER with a view to their future implementation in DEMO. Because of this, a vast amount of reviews can be found regarding each of them [65,66,67,68,69,70,71,72].

#### 5.1.2. Advanced RAFM

In parallel to the validation and planning steps for the use of F82H and EUROFER97 in ITER, there are also ongoing developments to modify these steels and improve their performance for DEMO. Specifically, a new generation of 9% Cr steels is being developed, known as advanced RAFM [73]. The strategic vision is to retain the basic structure and advantages of RAFM steels while improving their operational performance. Some target requirements are to operate in a higher temperature range and to be able to withstand neutron damage up to 70 dpa (with the possibility of extending this target to about 150 dpa), while keeping their reduced activation properties [11,23,60,74]. Within EUROfusion, two clear goals have been adopted for EUROFER97:DBTT reduction with the objective of using it in water-cooled designs.Enhancing its resistance to high temperatures, in particular to improve creep strength, with the objective of being used in He-cooled designs.

Modifications are also being made to F82H with subtle changes in chemical and thermodynamic processing, including:Limiting the amount of Ti to avoid loss of toughness.Increasing the Ta and N ratios to reduce radiation-induced embrittlement and improve creep resistance, respectively [74].

### 5.2. SiC/SiC

SiC/SiC composites have experienced a major development for nuclear fusion applications in recent decades. These materials consist of SiC fibers that are embedded in a high-crystallinity SiC matrix with a carbon or carbon/SiC multilayer interface.

These composites are generally manufactured by CVI (Figure 11) and have been shown to be resistant to neutron irradiation at elevated temperatures in terms of retention of mechanical properties, which is why this material is one of the leading candidates for nuclear applications. In addition, SiC itself has certain advantages over other candidates, including high temperature resistance (∼1600 °C), low neutron absorption, low activation and excellent chemical stability [56,75].

SiC is a brittle material but with a great facility to improve its fracture toughness by modifying the fiber, matrix and interface. It is composed of tetrahedrons of carbon and silicon atoms with strong bonds in the crystal lattice, which produces a very hard and tough material [55,76,77].

Figure 12 represents the predicted radioactivity of EUROFER and SiC/SiC samples irradiated in a fusion reactor after 25 years of operation at full power. After 100 years, its activity has been reduced by a factor of almost $10^{6}$. Therefore, it can be deduced that it is an excellent low-activation material [14].

The system composed of a silicon carbide matrix reinforced with silicon carbide fiber (SiC*_f_*/SiC) has undergone continuous evolution thanks to its great performance in hostile irradiation environments. This is why it is being considered for use in a variety of areas, both for structural fusion applications and for other high performance applications such as aerospace engineering.

#### 5.2.1. Flow Channel Insert

Among all available TBS designs (Table 3), SiC is of vital importance for the DCLL. In this design, a liquid PbLi alloy flows through a series of channels acting as a coolant and T breeder. As a result, it reaches high temperatures (∼700 °C) and provides high thermal efficiencies (∼45%). However, the development of this concept requires overcoming a high level of R&D in various areas of study [78,79].

Among the challenges to be addressed is the development of such channels, called flow channel inserts (FCIs). These are hollow square channels (∼5 mm) that contain the liquid metal flowing at 10 cm/s. Their main objectives are:To protect against corrosion and/or infiltration of PbLi during its operation time.To provide thermal insulation in order to protect the steel structure (RAFM) of the blanket from the high temperatures of the PbLi.To electrically isolate in order to avoid EM interactions generated between the fluid and the intense magnetic field present in the reactor. This minimizes MHD pressure drop.

One possibility is the development of a sandwich-type material that includes a SiC porous core (thermal and electrical insulation) and a dense SiC coating (protection against PbLi corrosion and infiltration) [75,78,79,80].

Figure 13 shows the different geometries of the FCI prototypes produced in [79] by the gel casting method after being sintered, oxidized and CVD-SiC coated. All of them have a section of 25 × 25 mm^2^ and a porous core of 5 mm thickness.

Precisely, the method used to create these prototypes, called gel casting, is an advanced ceramic composite preparation technique developed at ORNL [81]. This method presents numerous advantages over conventional techniques used in PM, especially for the production of FCI, where the fabrication of complex shapes with a relatively large size is required.

The gel casting process is a low-industrial-cost technique with the ability to produce uniform bodies with high strength and with the possibility of reducing or limiting some defects such as particle agglomeration, pores and cracks [79,81,82]. This technology paves a new way for the preparation of ceramic parts with potentially nuclear applications.

#### 5.2.2. Tritium Permeation Barriers

The permeation of T through structural materials is a crucial issue for both radioactive safety and the reproduction of this element (TBR >1). Structural materials such as 316L and F82H steels cannot meet the service requirements of D-T fusion power plants because the solubility and permeation rate of H in these materials is quite large, particularly at high temperatures, which has serious consequences in terms of embrittlement. Such a reduction in steel permeation can be achieved by making use of tritium permeation barriers (TPB) [83,84].

Implicit in these barriers are high performance requirements such as radiation and corrosion resistance, low activity, high thermal mechanical integrity, breeder compatibility and applicability to large components [83].

The materials available for TPB can be classified into metal oxides, especially Al_2_O_3_, and non-oxide ceramic composites, such as SiC coatings [85].

In one of the designs of the TBM program (HCPB), for which the use of Li_4_SiO_4_ and/or Li_2_TiO_3_ as breeders is foreseen, certain studies [84] have proposed the use of SiC coatings due to the fact that they:Satisfactorily withstand corrosion tests.Reduce the permeability of steel by up to three orders of magnitude.Have an inert character, so they do not react with the surrounding medium, keeping the Li out of the EUROFER.

#### 5.2.3. Plasma Facing Components

Numerous PFC concept studies are currently underway due to the good compatibility between W-SiC (in terms of thermal expansion, bonding technology and operating temperature window) [86,87]. The development of technology to join SiC/SiC with itself or with other materials is essential for their integration into various nuclear applications. These bonds are intended to provide mechanical robustness, tolerance to neutron irradiation and a certain chemical stability in the operating environment [80]. When W is reinforced with a ceramic material such as silicon carbide, a so-called metal matrix composite is produced. SiC-reinforced W metal matrix composites have been fabricated and shown to present favorable properties, such as increased resistance to corrosion and abrasion.

Following the Fukushima disaster, SiC-based cladding was proposed to replace the current zircaloy and is also one of the leading candidates for use as a structural protective layer for fuel particles in Gen IV reactors. This research has significantly advanced fabrication technology as well as the understanding of material properties. Due to common technological hurdles, these technologies and insights can be applied to the development of fusion materials [75,76].

#### 5.2.4. Main Disadvantages

As a PFC, SiC will experience high heat fluxes and damage from neutrons and charged particles. These α particles will impact the surface of the material, resulting in a variation in the amount of He. It is therefore pertinent to try to further understand the behavior of SiC under fusion-relevant conditions (He-appm, dpa and irradiation temperature). Since current research is mainly focused on microstructural characterization, knowledge of transmutation effects on macroscopic properties will be essential to evaluate the performance of SiC in fusion reactors [75,76].

In [80], neutron irradiation of the fusion spectrum has been reproduced to test whether it is indeed one of the key factors limiting the lifetime of SiC compounds. Production rates of 50–180 appm He/dpa and 20–70 appm H/dpa are predicted for gaseous transmutations and 10–45 appm Mg/dpa, 5–18 appm Be/dpa, 3–14 appm Al/dpa and 0.2–1.5 appm P/dpa, depending on the blanket concept. As with other materials, the high rate of He production tends to cause swelling due to the stabilization of the interstitial helium at intermediate temperatures during ion irradiation and of vacancy clusters at high temperatures (>1000 °C).

One of the main obstacles today is the price of the components as well as the high compatibility to be achieved between the matrix, fiber and interface. In turn, the swelling of cavities in the application temperature window or the effects of large amounts of He at high temperatures are still unknown. Finally, one of the most commonly used fabrication processes (CVI) produces a microstructure that has approximately 10% porosity and is therefore permeable to gases. For these reasons, it has certain limitations such as a relatively low thermal conductivity and stress limit. As with vanadium alloys, there is some concern about the lack of manufacturing infrastructure and potential costs. There is therefore a need to develop more efficient fabrication methods and a large-scale joining technology [55,56,80].

### 5.3. Vanadium Alloys

Since they were considered as candidates for LMFBR cladding materials in the 1970s, V alloys have always been linked to the nuclear industry. In the 1980s, their use in fusion reactors was considered due to their low activation properties, and since then they have evolved to such an extent that they are now considered one of the three most promising structural materials for fusion reactors together with RAFMs and SiC/SiC composites.

Vanadium alloys play a key role, as they are contemplated for most advanced DEMO designs using liquid Li as the breeding and cooling material (Figure 14) due to their good compatibility with this element [88,89,90,91,92].

In addition, these alloys exhibit good resistance to corrosion and irradiation swelling and, in particular, maintain high temperature resistance. With a V alloy structure, blanket designs using liquid Li can increase the coolant temperature and achieve relatively high TBR values without the need to introduce Be as a neutron multiplier. This has a very attractive consequence, namely that without Be, the system is freed from the radiological problems arising from its toxicity. Nevertheless, this concept has two main problems: the T recovery of liquid lithium and the MHD pressure drop [90,93].

The typical vanadium-based alloy is V-4Cr-4Ti. The addition of Cr provides improved strength and creep resistance, while Ti provides good resistance to irradiation-induced void swelling in the vanadium matrix (BCC structure). During the fabrication and processing of these alloys, impurity levels (e.g., C, O and N) must be carefully controlled due to possible degradation of mechanical properties. Based on these considerations, a protective or high-vacuum atmosphere is normally used to prepare the alloy, whose maximum operating temperature is 700 °C [58,91,93].

#### 5.3.1. Main Disadvantages

The main drawbacks of this type of advanced materials stem from their poor large-scale manufacturing technology. To this, the detrimental effects of He transmutation on mechanical properties and the effects of radiation on fracture properties must be added. Again, in order to carry out a conclusive characterization of all these aspects, the development of a facility equipped with a fusion neutron spectrum source is essential.

The T retention characteristics of vanadium alloys leave much to be desired, as they possess a H permeability of at least two orders of magnitude more than any other blanket material and can form detrimental hydrides [88,94]. In addition, their high diffusivity and solubility coefficients create a serious problem as the embrittlement of the materials by H is something to be avoided at all costs since it contributes to their degradation [89,91].

V-4Cr-4Ti is expected to accumulate a damage level of 50–80 dpa/fpy, which may result in the formation of defects (dislocations, bubbles, etc). Possible effects of neutron damage include hardening, embrittlement or swelling.

The study of irradiation in structural materials indicates that V alloys show significant resistance to irradiation damage above 400 °C. However, their hardening below 400 °C is related to the formation of point defects and clusters. Additionally, it has been found that the dissolution of Ti-rich precipitates may affect the hardness of the welded joint [93].

The hardening of these alloys occurs significantly at relatively low temperatures, as illustrated in Figure 15.

#### 5.3.2. Near Future

As an advanced choice of structural materials for fusion applications, the manufacturing technology of V alloys has made great advances in recent years. Research on coating and corrosion, irradiation damage or H isotope retention has also progressed considerably. However, critical problems related to high temperature operation and low temperature embrittlement of the material remain to be solved [90,93].

Because efforts in recent years have focused more on a mature candidate such as RAFM, progress in the development of V-alloys slowed down as compared to a decade ago. Despite this and because advanced options need to be explored in order to mitigate risk and provide a higher long-term performance option, research into these materials is crucial and can be achieved through efficient use of the available infrastructure.

From this point of view, exploring new advanced materials to improve their performance becomes more meaningful. One example may the coating of V alloys in the FW with shielding materials, such as W layers, by a vacuum plasma spraying method (VPS) [90].

Finally, for a near future development it is worth mentioning the effort being made in assessing high-entropy alloys (HEAs). By creating HEAs from elements with favourable properties in terms of nuclear activation, materials that can withstand the nuclear fusion environment can be manufactured while minimising the radioactive waste produced. Such a material could be used in the extreme thermal and irradiation conditions of a fusion blanket as they demonstrate impressive combinations of attractive properties such as high strength and high fracture toughness. In addition, these alloys may demonstrate superior irradiation damage tolerance due to high lattice distortion and sluggish diffusion from multiple principal elements in HEAs [96,97,98].

These alloys, which are multi-element, equiatomic metallic systems, can crystallize in a single phase despite having high concentrations (20–25 atomic percent) of various elements with different crystal structures [99]. Fascinating new materials may emerge as this subject develops. Nevertheless, there have been no studies done on FCC alloys containing V, and theoretical conclusions to date have only been drawn on a small range of alloys [97].

### 5.4. Oxide Dispersion Strengthened Steels

As mentioned in Section 4.1, RAFM steels have been chosen as the structural material for the ITER blanket modules. However, these steels present certain limitations for future advanced DEMO systems. It is for this purpose that ODS steels may come to play a key role. The properties of these materials are based on a broad dispersion of oxides distributed almost homogeneously throughout the composite. These precipitates tend to be stable under high temperature conditions (∼900 °C) and practically chemically inert, providing a high performance material for nuclear applications.

ODS are potential candidates for fuel cladding in SFR, as well as for Gen IV fission and fusion reactors. They consist of a series of fine oxide particles in the RAFM steel matrix, resulting in the trapping of irradiation-induced defects. The oxide particles also act as obstacles to the movement of dislocations, thus causing the strengthening of the steel at high temperatures. The size of the *dispersoids* (a substance that is dispersed in the form of microscopic particles in a medium that can be gaseous, liquid or solid) that impart the necessary properties to the matrix are in the nm range.

There is currently a wide variety of ODS, which can be classified into 3 groups: martensitic (9% Cr), ferritic/martensitic (12% Cr) and ferritic (14–15% Cr) [100,101].

#### 5.4.1. Main Constituents

Due to their structural variability, ODS steels have the following constituents, among others:**Y_2_O_3_**: This is the most important component, as it improves creep resistance at high temperatures by pinning mobile dislocations and delays the swelling of voids by acting as sinks for point defects produced during irradiation. It is optimized at 0.35%.**Cr**: Sets the maximum service temperature. Generally, steels whose Cr content is restricted to 8–9% work below 600 °C, while those with Cr above 12% operate at around 800 °C. Finally, those with a content between 14 and 16% can achieve higher temperatures but also show embrittlement due to thermal aging.**N**: Of great use in ferritic ODS, its content is restricted to 0.01%, thus avoiding the formation of harmful TiN compounds.**Ar**: Its strict control (<0.002%) is essential to avoid embrittlement due to the formation of Ar bubbles during irradiation.**W**: Stabilizes ferrite and can reduce toughness. It is optimized to 2.0%.**Ti**: Refines Y_2_O_3_ particles (20 nm after MA) to ultrafine particles (2–3 nm). The beneficial effect of titanium saturates about 0.2% [102].

#### 5.4.2. Compatibility between RAFM and ODS Steels

The combination of RAFM and ODS steels can be very useful in expanding the design scope of the blanket modules, where ODS are placed in more severe environments and RAFM are used for large components in less demanding environments. The attractiveness of ODS is not only due to the oxide particles but also to the ability to control their structural morphology depending on the properties required. The fine particle distribution of Y_2_O_3_—the most frequent dispersoid—is essential to improve the high temperature strength of ODS steels [100]. This improvement is achieved by the dissociation of these particles by the MA process, which gives the material an ultrafine microstructure that provides unique properties [103].

Several studies [100] have shown that ODS steels have a higher operating performance than RAFM steels (Table 4):

Their promising properties make ODS one of the most promising structural materials for the future. Their compatibility of operation between fission and fusion environments makes it possible to unify research and resources. Thus, its potential applications are focused on GenIV fission reactors and as a potential substitute—or companion—for RAFMs in DEMO [100,101].

Specifically, its use in DEMO is proposed for the DCLL blanket design. This concept has limitations of use due to the maximum acceptable temperature at the FW and the compatibility of the structural material with LiPb, which limits the allowable interface temperature to about 550 °C. The use of ODS with a temperature limit based on higher strength would increase operability, but it should not be forgotten that the welding requirements would make fabrication difficult [104].

Numerous studies are currently underway to achieve better compatibility between RAFM and ODS. Specifically, a new oxide dispersion-strengthened EUROFER steel has been tested using a two-step MA route. Starting from atomized EUROFER powder, various particles with average sizes of ∼60 μm and ∼120 μm were separately milled after addition of Ti and Y_2_O_3_ at the nanometer scale. The final result has been a material with a microstructure characterized by two distinct regions: zones with high-density particles (HDPZ) and zones with low-density particles (LDPZ). The coexistence of these regions has been shown to significantly improve the mechanical properties of the new EUROFER-ODS compared to other equivalent steels [105].

#### 5.4.3. Problems to Be Solved

The biggest challenge presented by these materials seems to be their *anisotropy* (the quality of exhibiting properties with different values when measured along axes in different directions) as well as the difficulty in uniformly distributing the dispersoids and being able to control their stability under irradiation [100,101].

Recently, some studies have appeared [57] that show the worrying tendency of these steels to retain D. Specifically, the diffusivity of D in ODS is an order of magnitude lower than in RAFM and CNA steels and, consequently, the effective solubility of D is 2 to 10 times higher.

TDS measurements were carried out to evaluate the deuterium *desorption* (the emission of a fluid previously adsorbed by a material) of these materials and, after applying a static thermal loading of deuterium at 723 K for 1 h under a pressure of $1.0\times 10^{5}$ Pa, it could be seen that ODS steels exhibit the highest D retention and have broader desorption peaks, indicating the presence of various capture sites due to the existence of ultrafine grains and high-density oxide nanoparticles characteristic of ODS [57]. Moreover, their complex production by MA results in high fabrication cost as well as low production volume and the aforementioned high probability of anisotropy of mechanical properties and low toughness [57].

## 6. Diagnostic Materials

Diagnostic systems will play a fundamental role in the control of the fusion process and will allow us to achieve a better understanding of plasma physics. To accomplish this, the tokamak must be equipped with sensors and instrumentation to fully explore the operating environment [106]. For the continued operation of ITER, it is of vital importance to predict the behavior of structural and functional materials under neutron and γ-particle radiation, as these will degrade the material properties and thus deteriorate its performance.

There are about two hundred different diagnostic systems in ITER located at various locations in the machine, ranging from the FW to the outer areas of the VV. The greatest radiation damage will be induced in the components closest to the plasma, i.e., the FWS and retroreflectors (RR) [107].

Moving away from the FW, several magnetic sensors and *bolometers* (an instrument that measures the total amount of electromagnetic radiation coming from an object for all wavelengths) are placed between the blanket modules and the VV wall. Finally, the electronics, cameras and optical fibers are examples of external diagnostic components, also called “ex-vessel components”.

As can be seen in Figure 16, as the components are further away from the FW, the radiation effects they suffer are reduced. There are two types of effects on their performance:Dynamic radiation effects: Substantially influence the performance of components from the onset of exposure to radiation environments.Long-term radiation effects: Gradually degrade their performance capabilities [107].

Hundreds of diagnostics will be used in ITER, but the number and type will be reduced in DEMO due to the restricted space available for diagnostics and harsh operation conditions. This is so because the requirements for achieving high reliability in DEMO plasma control are much higher than in any other existing fusion device, since operational failures that could lead to disruptions and their damaging consequences in inner DEMO components must be strictly avoided.

Hence, as compared to ITER, the implementation of diagnostics at DEMO is even more limited due to adverse effects that degrade the front-end components (e.g., ionizing radiation or erosion and deposition on the material). To achieve high reliability and durability, the main diagnostic methods for DEMO have been selected based on their robustness, and the front-end components are intended to be mounted in protected locations in order to reduce loads to acceptable levels [108]. The limited space available and remote maintenance considerably reduce the design freedom for the layout of the control system and its main components. For this reason, the implementation of the diagnostic components in the blanket have to be observed and studied in depth [108,109].

At the same time, the need to retract components into protected locations can only be compensated by integrating a large number of individual channels and sightlines, which in turn represents a huge design effort and will occupy significant space in the tokamak [109].

Finally, it is important to note that not all key features of the DEMO plasma scenario and technology have yet been well-defined and simultaneously demonstrated in large-scale experiments under relevant conditions. Therefore, at the current stage of DEMO research, the development of diagnostic and control systems should generally proceed according to the significant uncertainties associated with the plasma scenario and the machine properties [108,109].

### 6.1. First Wall Samples and RetroReflectors

FWS and RR are part of the set of diagnostic materials that interact with the plasma and are located in the FWPs. The FWS are designed to monitor the sputtering of Be under neutral particle bombardment and its possible fuel retention, being able to reach operating temperatures of ∼300–400 °C. The samples have a thin layer of Be on the surface and are made of the structural material CuCrZr.

The sputtering resolution of Be in the range from 1 to 100 μm is obtained by special markers (5 to 10), which are placed in various depth ranges from the surface [11,107].

A candidate material for the marker is C, as it would form a strong chemical bond with beryllium (Be_2_C), thus reducing diffusion. A conceptual design of a FWS is shown in Figure 17, where the sample body allows the possibility of remote handling, while the reference point (made of Mo or W) indicates the deposition that has occurred on the sample.

On the other hand, RRs are optical devices that are part of the *polarimetry* (a technique that measures the optical rotation produced on a beam of polarized light passing through an optically active substance) diagnosis and reflect the light in a direction parallel to the incident beam. Their function is to return the sounding laser beam to the detectors located several tens of meters away. Temperature differences can introduce thermal distortions that modify the profile of the returning laser beam, introducing errors in the measurements [11].

The reflective parts of the RR are made of W, and the same structural material is considered for them as for the FWS. It should be added that both RR and FWS will have several parts made of type 316L steel. The expected service life of the FWS is expected to be approximately 2 years, while that of the RR will be less. Both elements should be able to withstand a neutron fluence of ∼$10^{24}$ n/m^2^, and the neutron doses (∼1 dpa) are not expected to significantly affect the thermal and mechanical properties of the materials used [107].

### 6.2. Mirrors

Optical components—mirrors, windows, lenses, etc.—are of vital importance for the operational control and safety of a nuclear fusion reactor. They are present in all the diagnostic systems required for the analysis of plasma optical radiation, so that almost half of the ITER operating parameters will be measured with this type of device [11].

Nevertheless, these components cannot be used directly facing the plasma. That is why every diagnostic system includes a reflective mirror, also called a first mirror (FM). Thus, optical designs for diagnostics in ITER include a plasma-facing FM in the high-radiation region, followed by at least one secondary mirror (SM) for the signal to finally reach the lenses. This process is depicted in Figure 18.

Single-crystalline materials (Rh and Mo) have demonstrated good optical performance under plasma sputtering conditions and are currently considered the leading materials for FMs [111].

Specifically, the main material of this FM is monocrystalline Mo, which is characterized by good sputtering resistance, thermal shock resistance and good compatibility with the RF wave mirror cleaning method. Therefore, it can cope with situations dominated by both erosion and co-deposition. Other materials that are also considered, but have not yet shown this compatibility in all aspects, are nanocrystal-coated polycrystalline Mo, W, Au or other high-reflectivity materials coated with a thin oxide film (Al_2_O_3_, ZrO_2_). The fluence expected for FMs is ∼$10^{23–24}$ n/m^2^ over the entire lifetime of ITER with temperatures that range from 50–300 °C, depending on the location. Stainless steel grade *316LN-IG* (a modification of grade 316LN with a reduced concentration of activation-susceptible elements (mainly Co, Nb and Ta) [112] ) is the main structural material [107,112,113].

### 6.3. Windows

In order to reach the desired temperature in the plasma, external heating methods are necessary. The question one may ask is: how does this extra energy get into the hermetically sealed VV and stay there?

The answer lies in using windows that act as a “transparent” but highly resistant barrier made of artificial diamonds. Specifically, ITER’s diagnostic systems will use more than 100 sets of windows in the primary and secondary vacuum boundaries [113]. F4E has signed a contract with a German company that will be responsible for the production of 60 diamond discs made by means of CVD. Each of these transparent barriers will be bonded to the window body through a thin metal foil [114].

*Diffusion welding* (a joining process by heat and pressure where the contact surfaces are joined by diffusion of atoms) (DFW)—either Al or Au—is considered the gold standard technique for the assembly of ITER windows [115]. Initially, candidate materials for the windows were fused silica, synthetic crystalline quartz and barium fluoride, among others [107].

The samples in Figure 19 will have a thickness of 1.1 mm and a diameter of 7 cm, but why did the scales finally tip in favor of diamond?

Windows are fairly common elements in X-ray or similar machines used in scientific facilities to act as a barrier. However, these more common types of windows are not prepared to withstand the conditions of ITER. For this reason, a search was initiated for a synthetic diamond window that would satisfy the required conditions, i.e., allow the corresponding microwave beam flux with the electron cyclotron heating system and at the same time protect the surrounding systems. In addition, these diamond blades not only have excellent mechanical and thermal properties, but also a considerable radiation hardness of up to approximately $10^{-4}$ dpa. This is of particular relevance as they must comply with strict radiological regulations since they will act as a barrier against T [113,116].

Despite all these positive aspects, it has also been shown that diamond exhibits a substantial drop in thermal conductivity under irradiation as a consequence of phonon scattering [117]. This has implications for its use in DEMO due to power transmission requirements and will require careful design to minimize exposure. While there are some initial tests of windows incorporating new materials, this field will need to be developed for higher fluences in the future. For example, amorphous SiO_2_ is currently the candidate for Vis-NIR-type windows, while for the millimeter-range IR-FIR, a selection of other materials is being investigated, such as CaF_2_, BaF_2_ or ZnS, among others [117].

### 6.4. Bolometers

Bolometric systems placed around the VV provide information on the spatial distribution of the radiated energy in the plasma and in the divertor region by a data tomography process. They can thus measure the total radiated power of the plasma with largely constant sensitivity from the visible range up to a photon energy of ∼25 keV [118,119].

A bolometer consists of a heat-absorbing body connected to a sink (an object that is kept at a constant temperature) through an insulating material or substrate. They were invented about 150 years ago and have since been used in various branches of physics (e.g., in astronomy to detect very slight changes in radiation). A basic bolometer consists of a thin strip of metal that absorbs radiation and is heated, after which the resistance of this strip is measured to determine how much radiation has been absorbed [118].

In the ITER bolometers (Figure 20), the absorber body is preferably a thin layer of gold up to 20 μm thick. The gold can transmute into mercury —in principle not a concern— and is also expected to improve its resistivity significantly during irradiation. The Au absorber body and Pt meanders will be applied on opposite sides of the substrate, which is made of silicon nitride [11,107].

In order to understand which materials should be used, numerous materials tests were carried out under irradiation which showed that bolometer sensors with a Si_3_N_4_ substrate can withstand irradiation doses corresponding to 0.3 dpa, a value above the 0.1 dpa limit imposed by the ITER project requirements, while their measured resistance after irradiation increased by only about 20–30 Ω [120].

The expected neutron fluence for bolometers is $5\times 10^{24}$ n/m^2^ with temperatures from 150 °C up to 250 °C during operation.

A total of 100 bolometers will be strategically placed around the tokamak to continuously measure the total radiation and radiation profiles, which can provide the plasma control system with much of the information necessary for its control.

One of the most demanding requirements for bolometers is that they must provide data readings in as little as 1 ms. In addition, they must be robust enough to operate in vacuum with high neutron fluxes and ambient temperatures above 200 °C. To obtain a complete picture of the profile of radiation, the arrangement of the bolometers is optimized to perform tomographic reconstruction using up to 500 lines of sight crossing the plasma. These lines are located in 22 cameras mounted directly on the surface of the VV distributed in six different sectors [11,118,119].

There is no need for this complete bolometric system during the early phases of ITER due to the low power scenarios that will be run at the beginning, so that there will not be much radiated power to measure. However, components that belong to the infrastructure, such as cables and mechanical supports, do need to be placed and aligned to be ready for the next assembly phase [119].

### 6.5. Wide-Angle Viewing System

WAVS is an optical diagnostic component intended to monitor the status of the FW and divertor for tokamak protection purposes. For this reason, it will provide real-time measurements in the visible (656 ± 1 nm) and infrared (4 ± 0.1 μm) spectrum coming from the VV to avoid any potential damage to the PFCs. The material used for all hardware is 316L stainless steel, except for the mirrors, which are made of *Zerodur* (an inorganic, non-porous glass-ceramic made of Si, Al and Li oxide, characterized by extremely low and homogeneous thermal expansion throughout the entire volume). It will be composed of 15 sight lines installed in four equatorial ports (nos. 3, 9, 12 and 17) that cover approximately 80% of the FW surface, with an approximate length of 10 m [121,122,123].

In Figure 21, the various components (which share an optical path) transfer the visible and infrared signals from the port-plug through the interspace and bioshield regions to the port-cell [121]. Basically, for each line of sight, the light emitted by the PFCs is collected and travels more than 10 m through a series of mirrors and lenses to the cameras located at the back end of the port-cell. In total, the WAVS system will include over 600 opto-mechanical components, in addition to other non-optical ancillary systems [123].

At the end of the system, and inside the shielded cabinet, a beam splitter fragments the wavebands into two independent channels, each having two identical chambers. Currently, it has already reached its preliminary design phase, and to reduce costs, its assembly will be modular (Figure 22) [121].

Due to the harsh conditions of ITER, exhaustive tests (irradiation, steam ingress…) have been carried out during the last years with collaborating entities (CIEMAT or KIT) with the aim of selecting suitable materials for these optical components [123].

## 7. IFMIF-DONES

For some years now, the EUROfusion program [124] has proposed to have a project for irradiation of materials that can simulate the neutron flux that they would undergo in a fusion reactor. The construction and operation of such a facility is now considered to be of vital importance for the future of ITER and DEMO. It was decided to go for a facility based on nuclear *stripping* (a process in which the core of a projectile grazes a target core which absorbs part of the projectile. The remainder of the initial projectile continues past the target) reactions and to include it in the project launched in the early 2000s known as IFMIF. The objective of this project is to deepen the knowledge of the behavior of the materials required for the construction of a future fusion reactor [125,126,127].

This is how the idea of IFMIF-DONES was born: a facility that will generate a high-intensity neutron source with characteristics similar to those of a future nuclear fusion reactor. Prior to this, another facility known as IFMIF-EVEDA had already been put into operation, and together with DONES, both make up the entire IFMIF project. EVEDA is located in Rokkasho, Japan and has been in operation since 2007, within the framework of the broader approach (BA) agreement between the EU and Japan. Its objective is to validate and provide actual operating data with a prototype partial DONES systems, i.e., it acts as a precursor facility. The decision to start construction of IFMIF-DONES is expected imminently [126].

### 7.1. Is This Project So Relevant?

DONES is a version aimed at characterizing structural materials. European in scope, coordinated by EUROfusion and F4E, it has been catalogued by ESFRI as a strategic research infrastructure for Europe, and Granada (Spain) has been proposed as its headquarters (Figure 23).

It will generate a neutron flux with a wide energy distribution covering the neutron spectrum typical of a fusion reactor (D-T). As is well known, ITER will be followed by the implementation of another fusion reactor, in this case a demonstration reactor (DEMO), which will allow the generation of electric power. For this energy production to be possible and profitable, it is necessary to develop materials capable of resisting high energy neutrons and high heat flux to be used in the FW and the blanket. Thus, testing materials and different blanket concepts in a fusion environment is an indispensable step for DEMO design [127].

Furthermore, since DONES will be available during the operation of ITER, the possibility that it could help this project in some aspects of its nuclear operation phase should not be ruled out [125]. PFMs in DEMO and future fusion power plants will be affected by an unprecedented flux of 14.1 MeV neutrons. The displacement and transmutation effects occurring in the FW due to He and H accumulation will limit the lifetime of components to only a few years at full power. Therefore, the replacement of its components should be carried out with a certain periodicity in order to avoid the dreaded embrittlement.

ITER will only present about 3 dpa in its full operational lifetime, while DEMO will present about 30 dpa/year at full power. DONES is aiming to deliver more than 20 dpa/year in its high-flux irradiation module or HFTM, which is capable of hosting about 1000 small samples [125,127,128].

DONES is a particle accelerator, specifically of *deuterons* (a deuteron designates the nucleus of the deuterium atom) accelerated to relatively low energies (∼40 MeV) but with a very high current (Figure 24). This particle beam will hit a 25 mm thick lithium target, so that the D-Li interaction generates a neutron nuclear reaction with energies similar to those faced by the FW of a nuclear fusion reactor [125].

The energy deposited on the Li target is very high, so it must be in a liquid state, running in a closed circuit known as a lithium loop. The technology used in DONES will be pioneering in many respects, largely due to the uniqueness of the accelerator, which is intended to utilize a current of 100–125 mA. At such a high intensity, the repulsion of the particles tends to cause the beam to expand and thus become more difficult to confine. To avoid this, the use of many magnets is necessary, which raises the component density well above the level of any other current accelerator.

Initially, the amount of energy to be managed in the DONES project is 5 MW, which will increase to 10 MW with the extension of the project to IFMIF. Due to the presence of neutrons, there are areas of the facility that will be significantly activated, and this implies that the maintenance of the core of the facility must be done remotely with the help of robots and automatic systems. Ensuring the safety and availability of the entire facility so that radiation times are relatively short is one of the biggest challenges of the project.

The neutrons to be generated are very special and have characteristics that allow the use of this facility not only in the field of nuclear fusion but also in physics and nuclear medicine studies [130]. At DONES, HFTM development will focus on irradiation on RAFM steels, W and copper alloys [128].

### 7.2. RAFM Steels Irradiation

Irradiation of RAFM steels (typically 8–10% Cr and 1–2% W) for application as a structural blanket material is considered a priority task for IFMIF-DONES. The irradiation capsules offer a volume of 54 cm^3^ for the samples and provide *quasi-thermal* (an isothermal process capable of indicating the temperature of the system at each step, requiring continuous thermal equilibrium) irradiation conditions. The irradiation temperature for each capsule varies between 250 and 550 °C. In central regions with considerable volumetric heating, samples will be embedded in liquid sodium to homogenize the temperature.

The incident neutron flux density in the HFTM is $5\times 10^{14}$ n·cm^s−2^^s−1^ while the average structural damage rate of the capsules reaches 12–25 dpa/fpy. The helium production rate is about 13 appm He/dpa and the hydrogen production rate is about 53 appm H/dpa. These ratios are fairly homogeneous throughout the area and are very similar to those expected in the DEMO FW.

The overall lifetime of the HFTM (1–2 years) is limited mainly by three factors: the structural damage that its external surface can withstand, the creep damage to the hermetically sealed capsules at high temperature, and the lifetime of the electrical heaters and other instrumentation [128].

### 7.3. Cu Alloys and W Irradiation

W and Cu alloys are currently considered as reference materials for FW and divertor (Figure 3). Tungsten has been proposed due to its high melting temperature and CuCrZr alloys due to their high thermal conductivity together with their good mechanical properties. However, their behavior under the extreme irradiation conditions expected in ITER and DEMO is still unknown.

DONES has been conceived as a plant similar to EVEDA but simplified to provide, in a reduced time scale and with a limited budget, basic information on the damage to these materials. The fast neutrons from the fusion reaction activate and damage the divertor and the blanket, so some of its components must be replaced periodically. The divertor impact area will be exposed during normal operating conditions to fluxes of up to 20 MW/m^2^ and even GW/m^2^ during transient events such as disruptions and ELMs.

Precipitation hardened CuCrZr alloys have been chosen as heat sink materials for the divertor and FW. This is due to their good ductility, high thermal and electrical conductivity and high commercial availability. In addition, they exhibit high fracture toughness and high resistance to radiation damage [131].

The temperature range of W in such applications spans from about the maximum temperatures reached by heat sinks (e.g., 550 °C for EUROFER and about 350 °C for copper) to almost the melting temperature of tungsten during short transient periods [128]. For temperatures up to 550 °C, irradiation of both materials can be performed in standard HFTM capsules [126].

For a dedicated W irradiation scenario, a structural damage rate of 1–3 dpa/fpy, a helium production rate of 9 to 10 appm He/dpa and a hydrogen production rate of 20 to 29 appm H/fpy can be achieved. On the CuCrZr side, structural damage rates of 5 to 30 dpa/fpy, a helium production rate of 6–8 appm He/dpa and a hydrogen production rate of 48–50 appm H/dpa can be achieved. Thus, for both W and CuCrZr, the values expected in DEMO are even surpassed [131].

The data in Table 5 show that the damage dose rate requirements called for in the fusion roadmap for Cu and tungsten alloys for DONES meet the maximum values calculated in the area of the DEMO divertor with DCLL design. The damage dose rate and the H and He production are analyzed at the different locations and compared with the real irradiation conditions in the FW and divertor [131].

These simulations (the experimental results presented were obtained with the *McDeLicious* code [131] used for neutron transport calculations) performed for the HFTM (Figure 25) are intended to evaluate whether the tungsten and CuCrZr alloys—planned to be used in the first and second layer of the DEMO divertor, respectively— meet the expected values for a DCLL type concept according to the damage requirements set out in the EUROfusion roadmap. The objective is to evaluate the amount of irradiated volume subjected to a given damage dose rate (dpa/fpy). Radiation damage limit values are thus obtained, which make it possible to identify the most favorable location for irradiating each material, taking into account the He and H ratios.

The W values chosen correspond to the area closest to the plasma surface, and those for CuZrCr alloys correspond to the region behind the W layer. The main conclusion is that, for both tungsten samples and Cu alloys, the established damage requirements (5 dpa for CuCrZr and 1 dpa for W) are achieved in most cases in the irradiation area. It can therefore be deduced that the DONES HFTM will be a suitable place to carry out the corresponding tests of this type of materials with a view to their subsequent use in DEMO.

Figure 26 reflects a number of interesting conclusions:

(i)The volume in the HFTM decreases the higher the damage dose.(ii)The maximum damage dose rates achieved are 5, 27 and 38 dpa/fpy for W, EUROFER and Cu alloys, respectively.(iii)The volumes that meet the damage requirements for all three materials are sufficiently high for the need of this type of experiments [131].

## 8. Conclusions

The problems of developing, testing, verifying and ultimately qualifying materials for the fusion reactor vessel environment are one of the great materials research challenges of recent times. The situation is very complex due in large part to the urgency of developing fusion reactors to meet the planet’s energy and environmental needs.

The development of ITER has been a giant step forward. Important measures that have been done to address important issues include the initiation of multiple irradiation campaigns, advanced high heat flux simulations, as well as the continued development of multi-scale models of materials. The lack of technologically ready materials is clearly a concern for DEMO, aiming to facilitate more effective planning and targeted materials development in line with EUROfusion’s strategic plans.

For the range of operating conditions expected in ITER and with even greater relevance for DEMO, a qualified database must be generated to demonstrate that candidate materials meet a series of indispensable design requirements. Throughout this paper, several materials have been analyzed, from the most studied and used today such as W, RAFM steels and Be, to the most promising ones for future projects such as SiC composites, V-alloys or ODS steels among others.

Although the materials for ITER are already defined, long-term development does not stop and hundreds of studies are being carried out to find new materials or manufacturing techniques that can be used in ITER to overcome the even more difficult and demanding conditions of DEMO. Numerous promising designs have been discarded, which shows the commitment of the nuclear industry to constantly strive for the highest possible degree of perfection and safety.

Technologies such as tritium generation have received special emphasis within the ITER R&D program, as it is expected to have a major impact in the future. For this reason, a review has been made of the different blanket designs that are being considered, as well as the materials that can contribute to a greater extent to a better efficiency.

Clearly, there is a strong overlap with nuclear fission materials research; however, fusion materials present a few additional challenges. Structural materials will suffer a severe operating environment, with damage from fast neutron irradiation, high temperature, and high stress. Requirements for these structural materials include low activation, good compatibility with various coolants, and irradiation resistance, among others. Thus far, three main candidates for low-activation structural materials have been proposed for the FW. These are —in order of relevance— RAFM steels, SiC/SiC ceramic composites and V-alloys.

For their part, diagnostic systems will play a key role in the control of the fusion process and will allow us to achieve a better understanding of plasma physics. To achieve this, the tokamak must be equipped with sensors and instrumentation to fully explore the operating environment.

Finally, one of the major limitations is related to the difficulty of reproducing a realistic fusion neutron spectrum to test candidate materials for DEMO. Fortunately, the development of IFMIF-DONES seems to solve this problem. In fact, some of the most advanced materials, such as Cu alloys, W or RAFM steels, will be studied in this facility.

## Figures and Tables

**Figure 1 materials-15-06591-f001:**
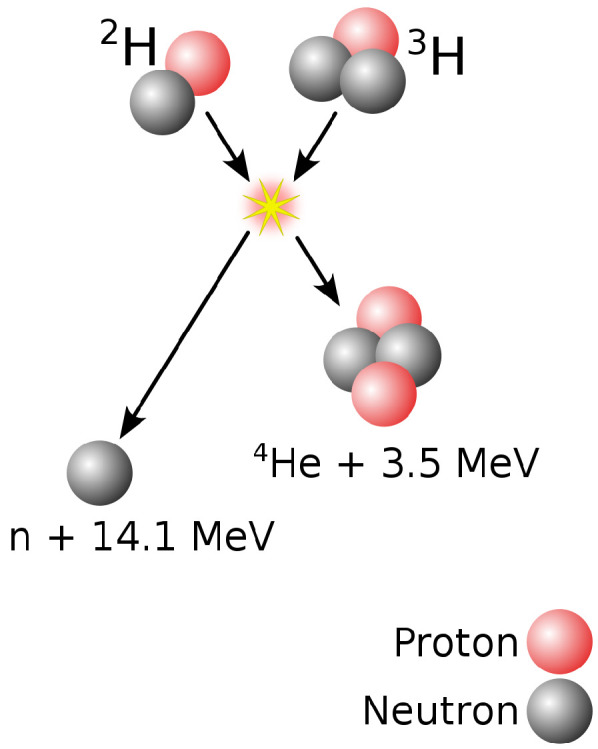
Nuclear fusion reaction. Reprinted from Wykis/WikimediaCommons.

**Figure 2 materials-15-06591-f002:**
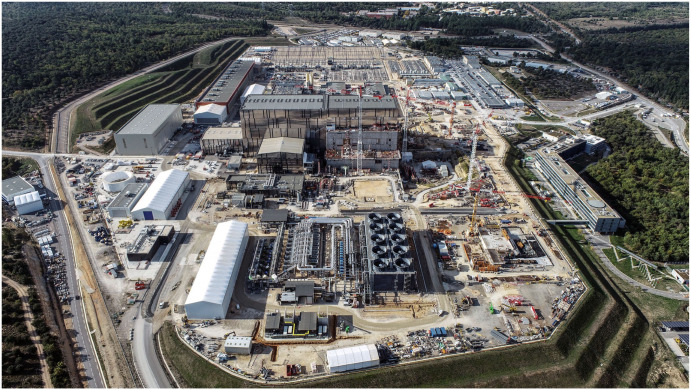
Aerial view of the ITER site. Reprinted from Ref. [5]. Credit © ITER Organization, 2022.

**Figure 3 materials-15-06591-f003:**
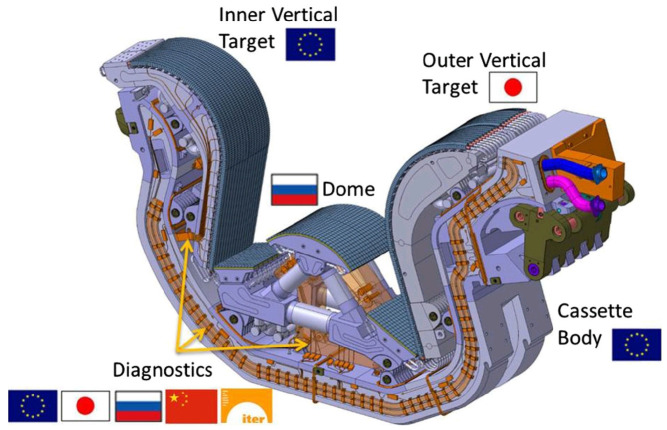
Parts of one of the 54 divertor modules. Reprinted with permission from Ref. [11]. Credit © ITER Organization, 2019.

**Figure 4 materials-15-06591-f004:**
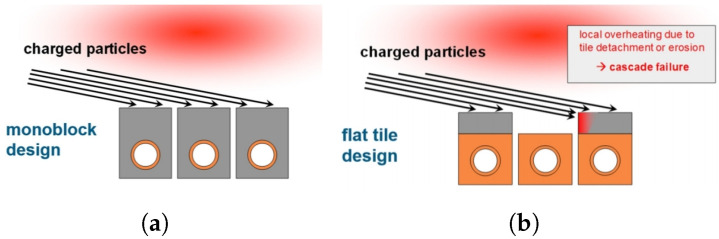
Main PFC designs for ITER. The charged particles depositing their energy are guided by the magnetic field lines. (**a**) Monoblock design. (**b**) Flat tile design. Reprinted from Ref. [12].

**Figure 5 materials-15-06591-f005:**
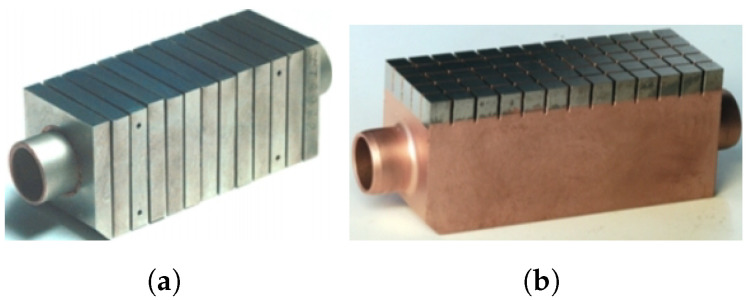
Prototypes of W designs as PFC for ITER. (**a**) Monoblock design. (**b**) Flat tile design. Reprinted from Ref. [12].

**Figure 6 materials-15-06591-f006:**
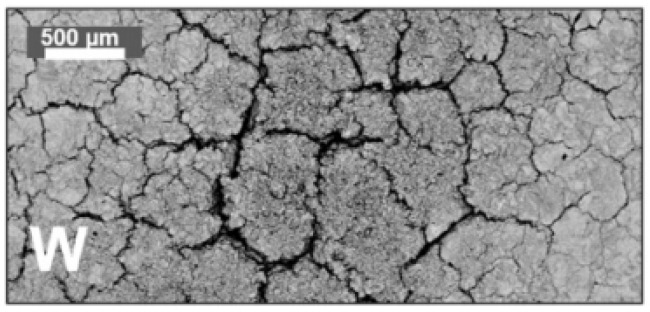
Surface damage with cracks induced by electron beam pulses. Reprinted from Ref. [12].

**Figure 7 materials-15-06591-f007:**
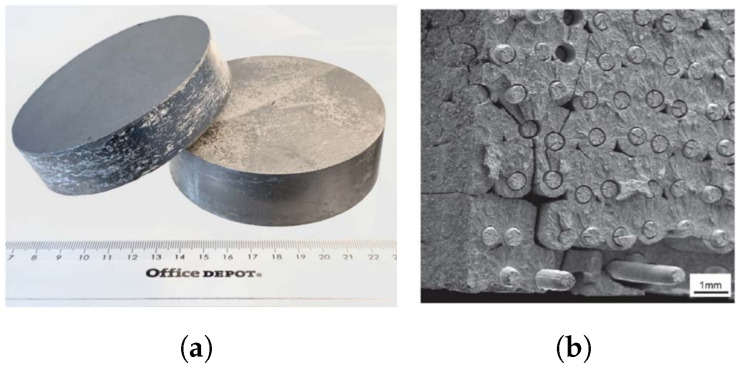
(**a**) PM W*_f_*/W prototype (**b**) typical fracture surface of CVD W*_f_*/W. Reprinted with permission from Ref. [29]. Credit © IOP Publishing, 2021.

**Figure 8 materials-15-06591-f008:**
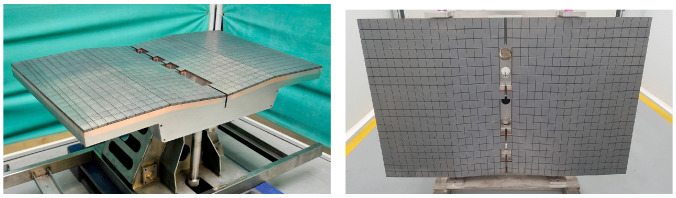
Prototype panel for the FW manufactured by Framatome. Reprinted from Ref. [39].

**Figure 9 materials-15-06591-f009:**
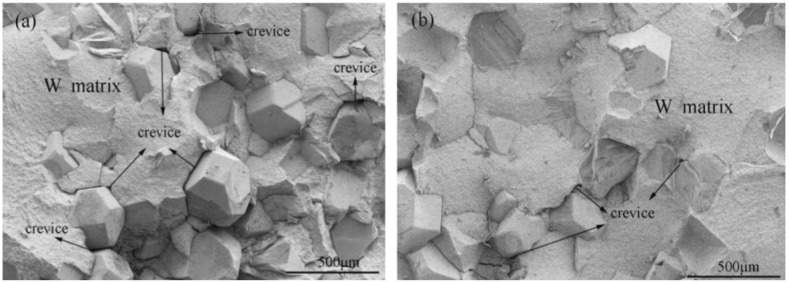
Microstructure of fractured surfaces: (**a**) uncoated and (**b**) coated. Reprinted with permission from Ref. [45]. Credit © Elsevier, 2019.

**Figure 10 materials-15-06591-f010:**
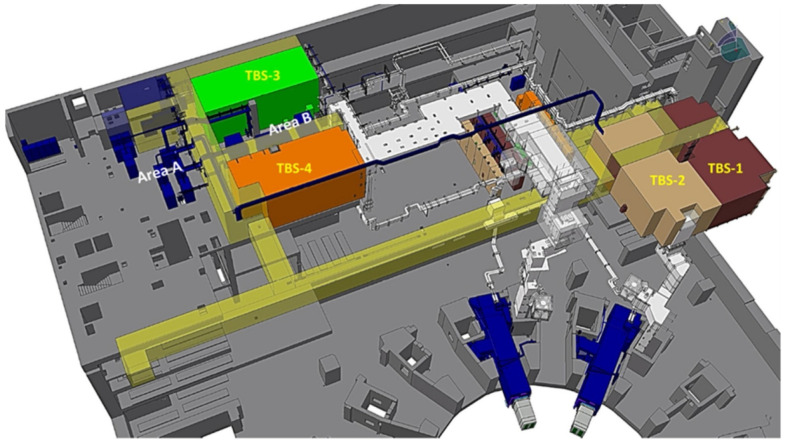
General view of the distribution of the main elements of the 4 TBS within the tokamak and tritium building. Reprinted with permission from Ref. [59]. Credit © Elsevier, 2020.

**Figure 11 materials-15-06591-f011:**
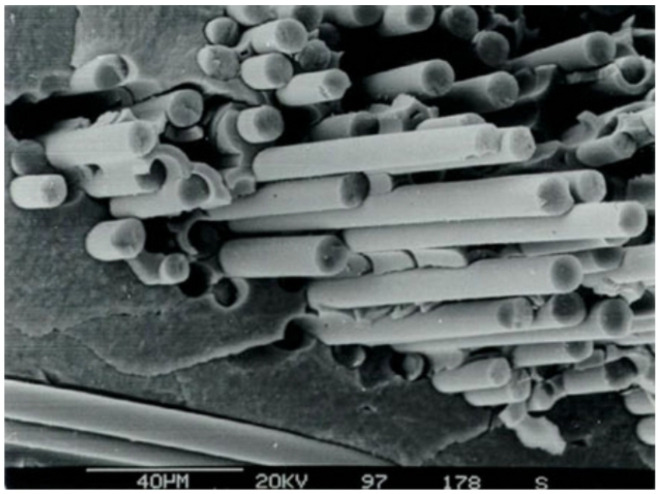
Microstructure of a CVI SiC/SiC composite. Reprinted from WikimediaCommons. Credit © MT Aerospace AG, 2006.

**Figure 12 materials-15-06591-f012:**
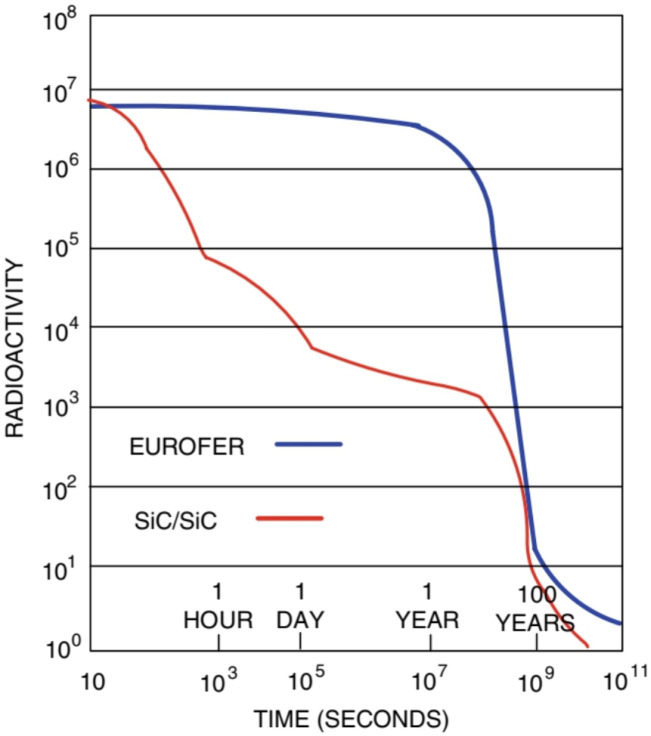
EUROFER and SiC/SiC radioactive decay in logarithmic scale. The vertical axis is in units of Bq/kg. Reprinted from Ref. [16].

**Figure 13 materials-15-06591-f013:**
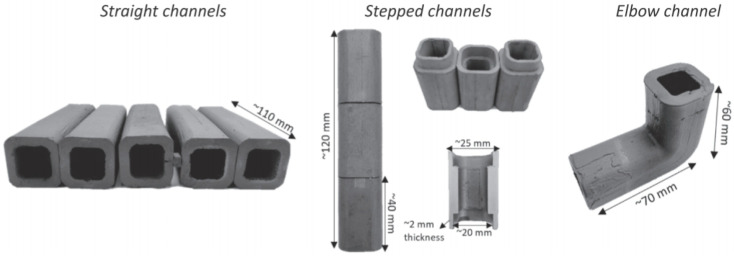
SIC-sandwich prototypes produced by gel casting at lab-scale. Reprinted from Ref. [79].

**Figure 14 materials-15-06591-f014:**
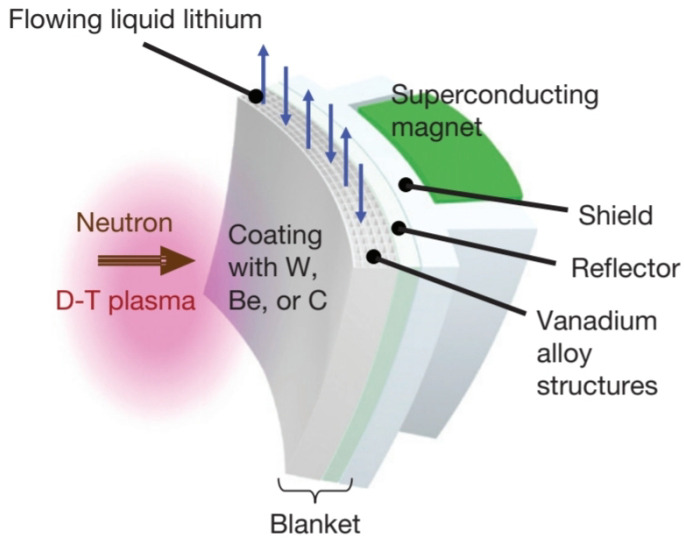
Illustration of a self-cooled Li blanket with structural material V-4Cr-4Ti. Reprinted with permission from Ref. [88]. Credit © Elsevier, 2012.

**Figure 15 materials-15-06591-f015:**
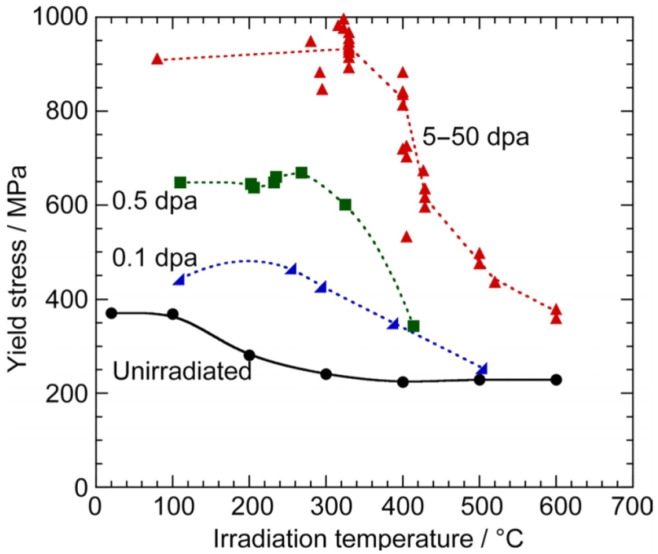
Yield strength of V alloy as a function of temperature and irradiation dose. Reprinted from Ref. [95]. Credit © Elsevier, 2019.

**Figure 16 materials-15-06591-f016:**
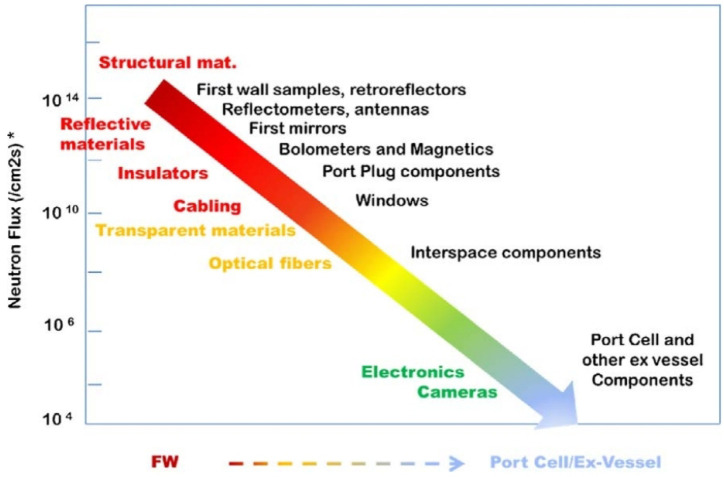
Neutron flux over the different diagnostic systems. Reprinted with permission from Ref. [107]. Credit © Elsevier, 2017.

**Figure 17 materials-15-06591-f017:**
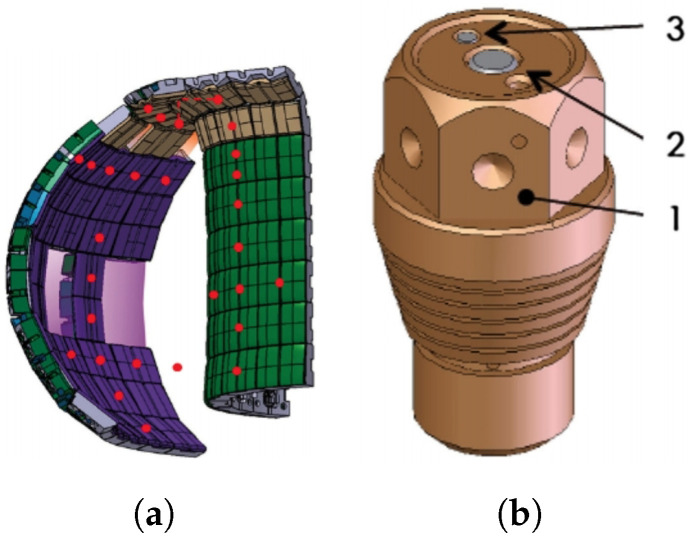
(**a**) Illustration of FWS distribution (red spots). (**b**) Conceptual design: 1. sample body, 2. Be insert with markers, 3. reference point. Reprinted with permission from Ref. [11]. Credit © ITER Organization, 2019.

**Figure 18 materials-15-06591-f018:**
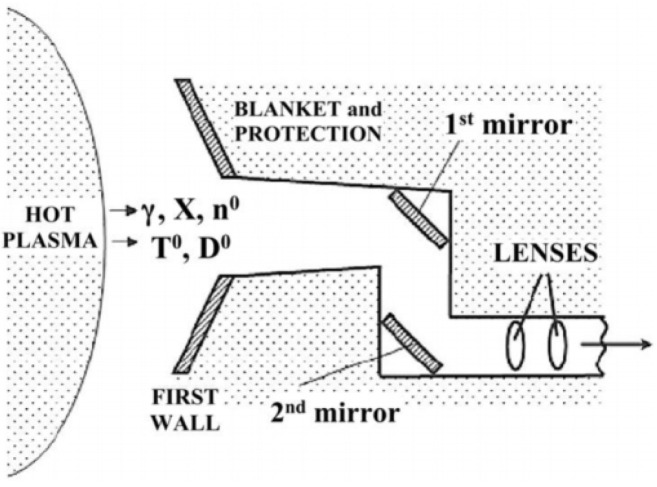
Schematic arrangement of the optical measurement components. Reprinted from Ref. [110].

**Figure 19 materials-15-06591-f019:**
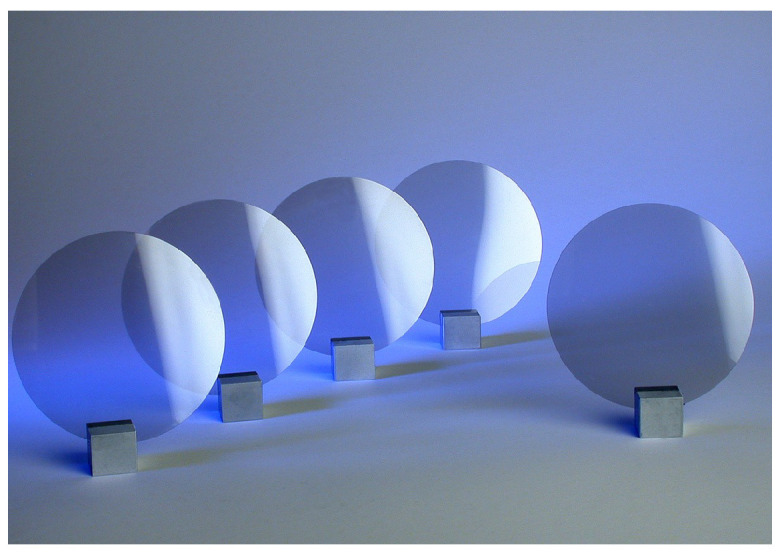
Sample of diamond windows produced by CVD. Reprinted with permission from Ref. [116]. Credit © Diamond Materials, 2020.

**Figure 20 materials-15-06591-f020:**
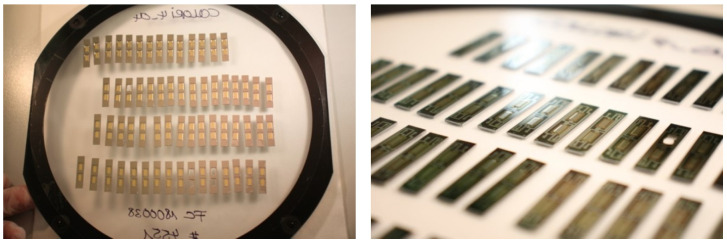
Prototype bolometer for ITER after receiving 10 thermal cycles at 400 °C. Reprinted from Ref. [118]. Credit © Stefan Schmitt, Fraunhofer-IMM, 2020.

**Figure 21 materials-15-06591-f021:**
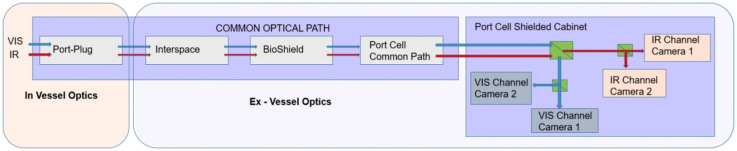
Block diagram of WAVS operation. Reprinted with permission from Ref. [121]. Credit © Elsevier, 2021.

**Figure 22 materials-15-06591-f022:**
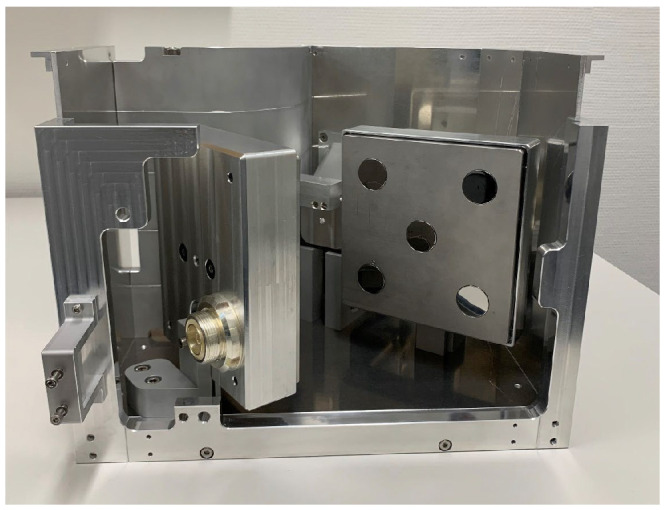
FM prototype with realistic WAVS geometry (scale 1:1). Reprinted from Ref. [123].

**Figure 23 materials-15-06591-f023:**
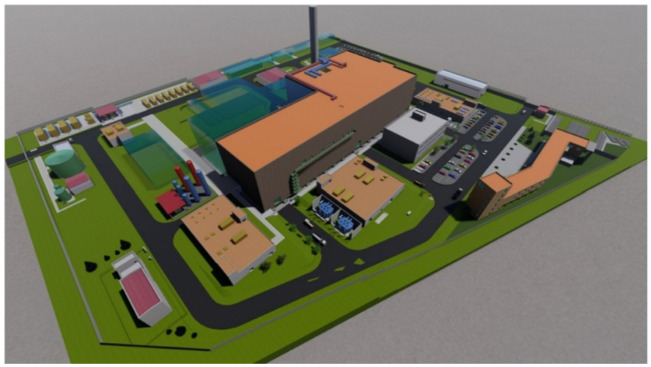
Recreation of the installation in its proposed location. Reprinted from Ref. [126].

**Figure 24 materials-15-06591-f024:**
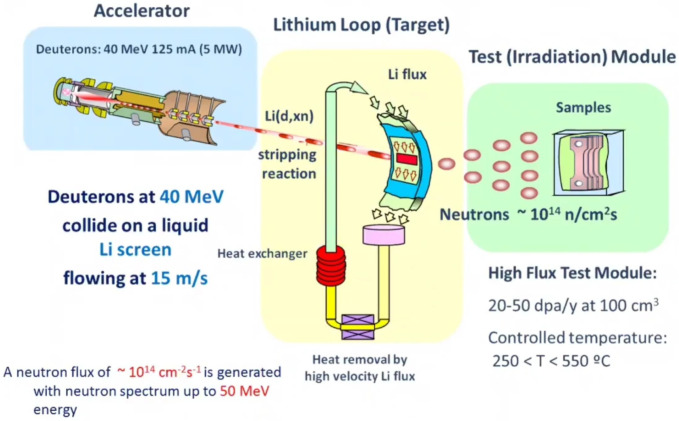
Process carried out in IFMIF-DONES. Reprinted from Ref. [129].

**Figure 25 materials-15-06591-f025:**
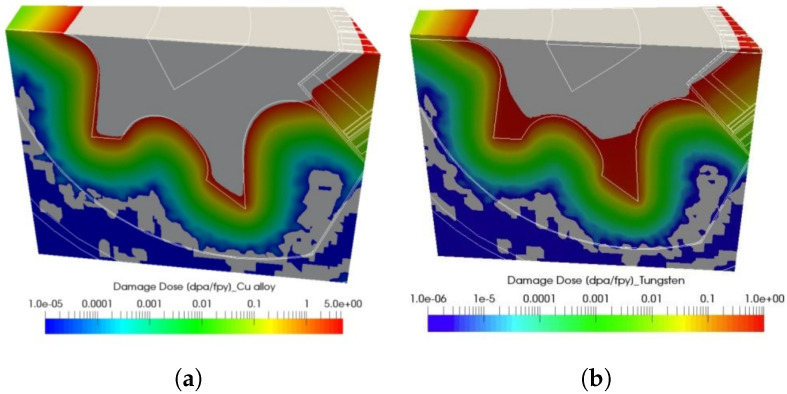
Simulation of irradiation damage in the divertor for (**a**) Cu alloy and (**b**) W. Reprinted with permission from Ref. [131]. Credit © IOP Science, 2017.

**Figure 26 materials-15-06591-f026:**
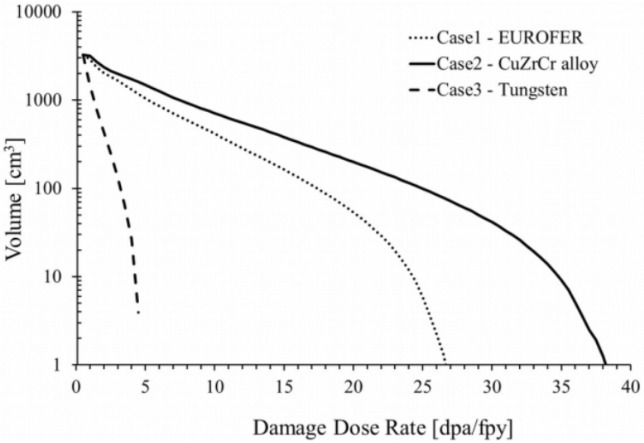
Damage dose rate as a function of HFTM volume with the 3 materials to be studied in DONES. Reprinted with permission from Ref. [131]. Credit © IOP Science, 2017.

**Table 1 materials-15-06591-t001:** Loads suffered by the PFCs in the FW. Adapted from Ref. [16].

	ITER	DEMO	PROTO	Units
Heat flux	0.3	0.5	0.5	MW/m^2^
Fusion power	0.5	2.5	5	GW
Neutron damage	<3	80	150	dpa
Neutron load	0.78	<2	2	MW/m^2^
Neutron load in life	0.07	8	15	MW-years/m^2^

**Table 2 materials-15-06591-t002:** Composition of F82H and EUROFER steels. Adapted from Ref. [16].

	Cr (%)	W (%)	V (%)	Ta (%)	C (%)
**F82H**	8.9	1	0.2	0.14	0.12
**EUROFER**	7.7	2	0.2	0.04	0.09

**Table 3 materials-15-06591-t003:** Composition of the nine main TBM designs. Six of them are **more developed**, and the remaining three have been **recently approved**.

	Meaning	Structure	Breeder	Multiplier	Coolant	Producer Country
**DCLL**	Dual-Coolant Lithium-Lead	RAFM	Pb16Li	Pb16Li	He and Pb16Li	US
**HCCB**	Helium-Cooled Ceramic Breeder	RAFM	Li_4_SiO_4_	Be pebbles	He	China
**HCCR**	Helium-Cooled Ceramic Reflector	RAFM	Li_2_TiO_3_	Be pebbles	He	Korea
**HCLL**	Helium-Cooled Lithium Lead	EUROFER97	Pb16Li	Pb16Li	He	EU
**HCPB**	Helium-Cooled Pebble-Bed	EUROFER97	Li_4_SiO_4_ or Li_2_TiO_3_	Be pebbles	He	EU
**HCSB**	Helium-Cooled Solid Breeder	RAFM	Li_2_TiO_3_	Be pebbles	He	India
**LLCB**	Lithium-Lead Ceramic Breeder	RAFM	Pb16Li or Li_2_TiO_3_	Pb16Li	He and Pb16Li	India
**WCCB**	Water-Cooled Ceramic Breeder	F82H	Li_2_TiO_3_	Be pebbles	H_2_O	Japan
**WCLL**	Water-Cooled Lithium Lead	EUROFER97	Pb16Li	Pb16Li	H_2_O	EU

**Table 4 materials-15-06591-t004:** Performance comparison between RAFM and ODS. Adapted from Ref. [100].

	Max. Temp. Allowable	Max. Stress. Allowable	Acceptable He Content	Corrosion Rate (Oxide Thickness)	Creep Rupture Time
**F82H**	539 °C	359 MPa	<600 appm (40 dpa)	0.6 mm	180 MPa; 550 °C
**15Cr-ODS**	780 °C	359 MPa	>1000 appm (65 dpa)	0.003 mm	180 MPa; 780 °C

**Table 5 materials-15-06591-t005:** Damage dose, production and ratio of H and He in the divertor. Adapted from Ref. [131]. Credit © IOP Science, 2017.

	Damage Dose Rate [dpa/fpy]	H Production [H appm/fpy]	He Production [He appm/fpy]	H Ratio [H appm/dpa]	He Ratio [He appm/dpa]
**CuCrZr**	5	180	30	36	6
**Tungsten**	1	8	5	8	5

## Data Availability

Not applicable.

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
