# Peer review of "Materials to Be Used in Future Magnetic Confinement Fusion Reactors: A Review"

_materials, 2022, doi:10.3390/ma15196591_

Round 1
Reviewer 1 Report
Report
Materials to be used in future MCF reactors: A review.
= = = = = = = = = = = = = = = = = = = = = = = = = = = = = = = = = = =
The authors review materials to be used in future nuclear fusion reactor, in particular in magnetic confinement reactors. The manuscript is well written and shall be useful so I may have only minor concerns with this paper.
First one.
I think the tittle should not contain an acronym.
A better tittle would be
Materials to be used in future Magnetic Confinement Fusion reactors: A review.
General comment
Some acronyms are used without explaining it before, for example TBR, for Tritium Breeding Ratio or later TBMs, just to name two examples.
Some are easy to infer but would be good if the most important and most used ones were introduced before using them.
Also, in the list of Abbreviations & Acronyms I do not see necessary to include the chemical symbols of the elements, like Al, Au, Cu etc
The topic is vast so is understandable that not all subtopics can be fully covered. However, some issues could be discussed further
In order of appearance:
3.4 Materials design requirements
I would add radiation resistance and nonactivation properties after irradiation.
4 Plasma Facing Materials
Some of the most serious damaging mechanisms to be considered in these materials are:
I would include two issues there:
1 Tritium retention
2 High velocity impacts of “dust” in the PFM
example
[] A. Fraile, P. Dwivedi, G. Bonny, T. Polcar, Nuclear Fusion (2022).
4.1 Tungsten
This review paper is worth citing
[] Makoto Fukuda et al. Thermal properties of pure tungsten and its alloys for fusion applications. Fusion Engineering and Design. Volume 132, July 2018, Pages 1-6
Note
4.1.1. ITER design
4.2.1. ITER design
Both subsections have the same tittle. Change that.
4.3 Diamond
Is diamond doping an economically attractive option? It seems expensive for the non-expert.
5.1 RAFM steels
I think the tittle in a section should not contain acronyms
Table 2. Composition of EUROFER and F82H steels. The rest % is supposed to be Fe?
The number of steels under investigation for nuclear technologies is quite a vast topic as well.
5.1.1. TBM Program
Idem. Would be better: Test Blanket Module program
This topic deserves a little bit more detail.
For example, if liquid PbLi is used, some issues arise, like liquid metal corrosion, or the behaviour of He and T in the liquid PbLi or the effects of magnetic fields in the fluid mechanics.
As there is a lot of literature in this area, and plenty of high-quality reviews, so some should be cited and the authors should try to summarize a few relevant experimental and theoretical results.
Some relevant papers include:
[] Alberto Fraile and Tomas Polcar 2020 Nucl. Fusion 60 046018. Volume and pressure of helium bubbles inside liquid Pb16Li. A molecular dynamics study.
5.2.1. FCI
Change tittle for > Flow Channels Insert
5.2.2. TPB
Change tittle for its meaning
5.2.3. PFC
Change tittle for its meaning
5.3.2. Near future
For a near future development is worth mentioning the effort being made in High Entropy Alloys.
6.1 FWS and RR
Change tittle for its meaning
Overall, the paper is interesting and shall be useful for researchers aiming to understand the state of the art in materials science for nuclear fusion technologies and the many challenges presented in the design of a nuclear fusion reactor.
Therefore, I believe it can be published after minor corrections.
Reviewer 2 Report
This paper presents an overview of the main materials to be used at ITER and DEMO class reactors and it is worthy of publication in this journal. But there are some points to be improved.
- Sections 1 - 3 contain too simple material that can be found in fusion text books or ITER documents, so they need to be re-written to focus on fusion environment, importance of fusion material, etc.
- In Sec. 4, it is better to compare conditions provided by ITER with what are required to test material in reactor conditions.
- Hundreds of diagnostics will be used in ITER but the number and type will be reduced in DEMO due to restricted space available for diagnostics and harsh operation conditions. It is necessary to add a paragraph addressing this point.
